# *Porphyromonas gingivalis* Outer Membrane Vesicles Promote Apoptosis via msRNA-Regulated DNA Methylation in Periodontitis

Ruyi Fan,[a,b,c] Yi Zhou,[a,b,c] Xu Chen,[a,b,c] Xianmei Zhong,[b,d] Fanzhen He,[a,b] Wenzao Peng,[a,b] Lu Li,[a,b,c] Xiaoqian Wang,[a,b,c] Yan Xu[a,b,c]

[a]Department of Periodontics, the Affiliated Stomatological Hospital of Nanjing Medical University, Nanjing, China
[b]Jiangsu Province Key Laboratory of Oral Diseases, Nanjing Medical University, Nanjing, China
[c]Jiangsu Province Engineering Research Center of Stomatological Translational Medicine, Nanjing, China
[d]Department of Periodontics, Taizhou Stomatological Hospital, Taizhou, China

**ABSTRACT** The outer membrane vesicles (OMVs) produced by *Porphyromonas gingivalis* contain a variety of bioactive molecules that may be involved in the progression of periodontitis. However, the participation of *P. gingivalis* OMVs in the development of periodontitis has not been elucidated. Here, we isolated *P. gingivalis* OMVs and confirmed their participation in periodontitis both *in vivo* and *in vitro*. Microcomputed tomography (micro-CT) and histological analysis showed that under stimulation with *P. gingivalis* OMVs, the alveolar bone of rats was significantly resorbed in vivo. We found that *P. gingivalis* OMVs were taken up by human periodontal ligament cells ([hPDLCs]) *in vitro*, which subsequently resulted in apoptosis and inflammatory cytokine release, which was accomplished by the microRNA-size small RNA (msRNA) sRNA45033 in the *P. gingivalis* OMVs. Through bioinformatics analysis and screening of target genes, chromobox 5 (CBX5) was identified as the downstream target of screened-out sRNA45033. Using a dual-luciferase reporter assay, overexpression, and knockdown methods, sRNA45033 was confirmed to target CBX5 to regulate hPDLC apoptosis. In addition, CUT&Tag (cleavage under targets and tagmentation) analysis confirmed the mechanism that CBX5 regulates apoptosis through the methylation of p53 DNA. Collectively, these findings indicate that the role of *P. gingivalis* OMVs is immunologically relevant and related to bacterial virulence during the development of periodontitis.

**IMPORTANCE** *P. gingivalis* is a bacterium often associated with periodontitis. This study demonstrates that (i) sRNA45033 in *P. gingivalis* OMVs targets CBX5, (ii) CBX5 regulates the methylation of p53 DNA and its expression, which is associated with apoptosis, and (iii) a novel mechanism of interaction between hosts and pathogens is mediated by OMVs in the occurrence of periodontitis.

**KEYWORDS** outer membrane vesicles, *Porphyromonas gingivalis*, periodontitis, apoptosis, DNA methylation

As the sixth most common human disease, chronic periodontitis is a major public health issue, with a high prevalence of 45% to 50% (1). The prevalent oral disease is usually characterized by chronic inflammation, which results from host inflammatory responses against Gram-negative anaerobic bacteria (2). An imbalance in host-bacterium interaction results in the disruption of periodontal homeostasis, which leads to the initiation and progression of periodontitis (3). The dysbiotic microbial environment triggered by the pathogen releases proinflammatory factors into the periodontium, which causes chronic inflammation and ultimately results in the destruction of soft tissue support, gingival recession, bone resorption, and the loss of teeth (4, 5).

The Gram-negative anaerobic bacterium *Porphyromonas gingivalis* is considered the

Address correspondence to Yan Xu, yanxu@njmu.edu.cn.

The authors declare no conflict of interest.

keystone pathogen in the development of periodontitis (6). *P. gingivalis* can colonize into the subgingival pocket to initiate the periodontitis (7), producing virulence factors, like gingipains and fimbrillin, to further damage the periodontal tissue (8), disrupt the balance of resident microbiota, and impair the host immune system (9).

Outer membrane vesicles (OMVs) are composed of a single lipid bilayer (which is derived from the bacterial outer membrane), which originates from all Gram-negative bacteria (10). *P. gingivalis* is also the most prolific OMV producer, and the sizes of the vesicles range from 50 to 400 nm (11). *P. gingivalis* OMVs contain virulence factors, like fimbriae, gingipains, and lipopolysaccharide (LPS) (12). *P. gingivalis* OMVs are highly inflammatory, with the ability to regulate neutrophils and macrophages and invade oral epithelial cells (13–15). These findings implicate *P. gingivalis* OMVs in important roles in periodontitis.

OMVs contains proteins, lipid, nucleic acids, and other biofunctional molecules, among which, small RNAs (sRNAs) in the OMVs have received attention for their potential gene regulatory functions as interspecies communication molecules (16, 17). There has been evidence that various bacteria utilize sRNAs interacting with RNA-induced silencing complex (RISC) to inhibit host immunity genes and promote intracellular survival (18–20). sRNAs enclosed and protected by OMVs could be transported into host cells and then regulate gene expression or immune responses (16, 21–23). Several bacteria, including periodontal pathogens (i.e., *P. gingivalis*, *Aggregatibacter actinomycetemcomitans*, and *Treponema denticola*) also produce a novel class of sRNAs of microRNA (miRNA) size (msRNAs) (17, 24, 25), which could be taken up by host cells, such as Jurkat T cells. Choi et al. demonstrated that after transfection of synthetic sRNA, immune responses were altered by decreasing levels of interleukin-5 (IL-5), IL-13, and IL-15 (17). In another study, the authors found sRNAs in *A. actinomycetemitans* OMVs increased tumor necrosis factor alpha (TNF-$\alpha$) via the Toll-like receptor 8 (TLR-8) and NF-$\kappa$B signaling pathways (26). sRNA23392 in the *P. gingivalis* OMVs was found targeting desmocollin-2 to promote oral squamous cell carcinoma migration and invasion (11). Bioinformatic analysis found that reads from *P. gingivalis*, *A. actinomycetemcomitans*, and *T. denticola* overlap some human genome regulatory regions and are aligned against some histone mark regions, which suggest possible function similar to long noncoding RNAs and epigenetic roles (27). However, extensive studies of OMVs and their sRNAs need to be performed to understand their roles, which could provide more insight into the development of diseases like periodontitis.

The eukaryotic genome constitutes transcriptionally active euchromatin and transcriptionally silent heterochromatin. Chromobox 5 (CBX5), also known as heterochromatin protein 1 alpha, is an architectural protein that binds DNA to form and maintain heterochromatin through association with the protein H3K9me3 (trimethylated of lysine 9 of histone H3) (28–30). Recent studies have found that CBX5 was related to heterochromatin-repressed inflammatory response, apoptosis, and death receptor signaling (31). CBX5 methylates H3K9 lysines, which form heterochromatin to silence relevant genes such as *IFNL1*, *CXCL10*, *CXCL11*, and *USP18* (32). Apoptosis is a programmed cell death process that responds to diverse stress situations. p53 is a crucial protein that induces apoptosis in response to various stimuli (33).

The participation of *P. gingivalis* OMVs in the development of periodontitis has not been elucidated. Here, we report that *P. gingivalis* OMVs were taken up by human periodontal ligament cells (hPDLCs), subsequently resulting in apoptosis and inflammatory cytokine release. This process was accomplished by the sRNA45033 in the *P. gingivalis* OMVs targeting CBX5. We also report that *P. gingivalis* OMVs induced apoptosis through p53 expression regulated by CBX5-associated H3K9me3. Therefore, our findings suggest that a novel mechanism of interaction between host, pathogens, and OMVs plays an important role in the occurrence of periodontitis.

## RESULTS

**Morphological characterization of *P. gingivalis* OMVs.** We collected OMVs from *P. gingivalis* culture supernatant, characterized the isolated *P. gingivalis* OMVs using

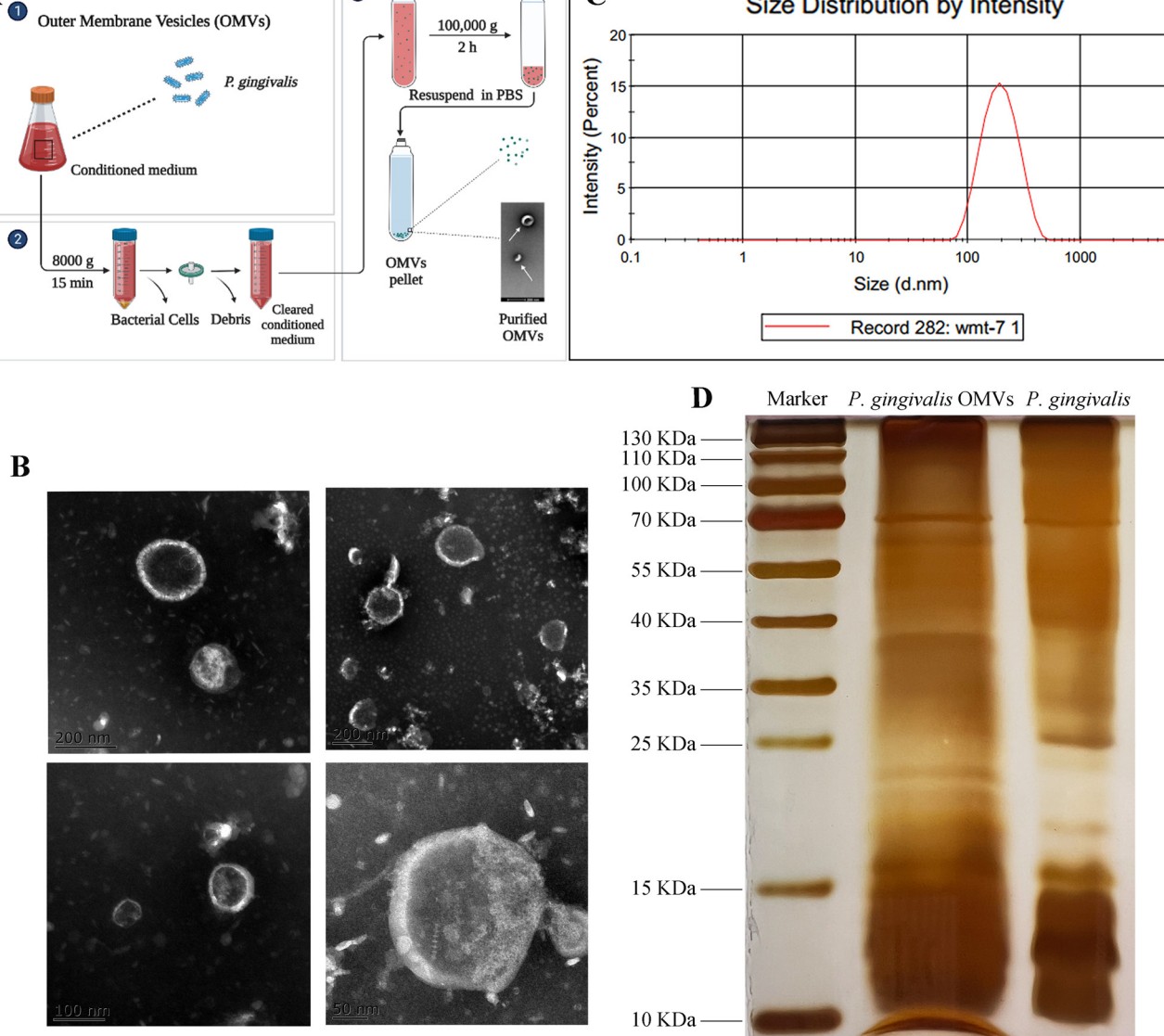

**FIG 1** Morphological characterization of *P. gingivalis* OMVs. (A) Illustration of OMV preparation. (B) The images of isolated *P. gingivalis* OMVs from culture supernatant were captured by TEM. Scale bars = 200 nm, 100 nm, and 50 nm. (C) The particle size distribution of isolated *P. gingivalis* OMVs was measured by dynamic light scattering. (D) The protein contents of *P. gingivalis* OMVs and *P. gingivalis* are shown by silver staining.

transmission electron microscopy (TEM), and measured their particle size distribution. The TEM images revealed that *P. gingivalis* OMVs are considerably round and oval (Fig. 1B), with a broad size range (Fig. 1C). The average diameter of the OMVs isolated from *P. gingivalis* cultures was 202.8 nm (Fig. 1C). Several protein bands of *P. gingivalis* OMVs were common to proteins of *P. gingivalis* according to silver staining (Fig. 1D) and Coomassie blue staining (see Fig. S1 in the supplemental material).

**P. gingivalis OMVs promote alveolar bone resorption *in vivo*.** To understand the role of *P. gingivalis* OMVs, a periodontitis model of Sprague-Dawley (SD) rats was constructed. A three-dimensional (3D) reconstruction image of the rat alveolar bone showed significant bone resorption in the *P. gingivalis* OMV-treated rats (OMVs group) compared to the control rats (CTRL group), as well as the rats with chronic periodontitis treated with *P. gingivalis* OMVs (CP+OMVs group) compared to the untreated rats with chronic periodontitis (CP model group) (Fig. 2B). The 3D reconstruction images and volumetric measurements from microcomputed tomography (micro-CT) analysis

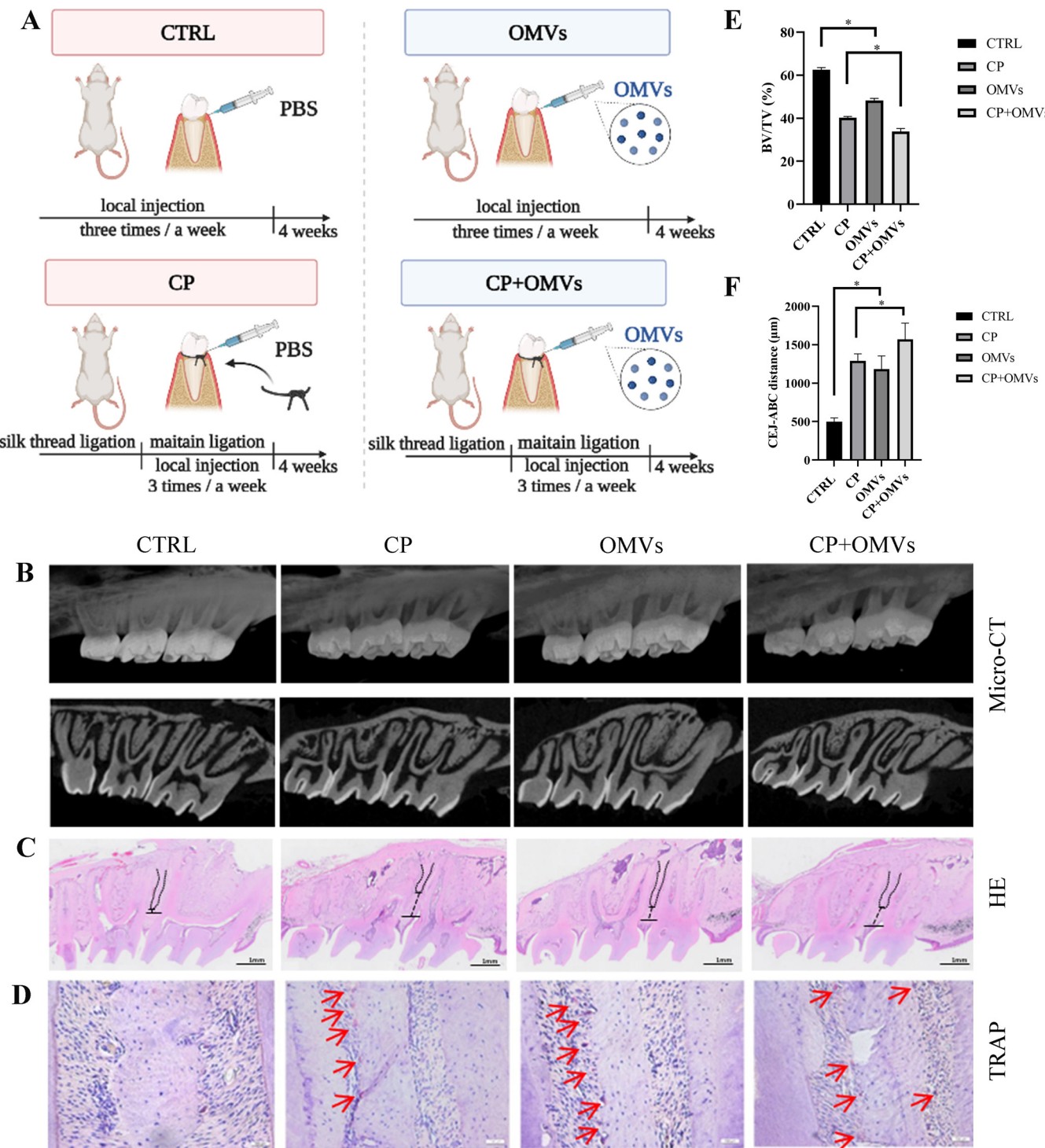

**FIG 2** *P. gingivalis* OMVs promote alveolar bone resorption *in vivo*. (A) Schematic diagram of animal experiments. (B) The level of the rat alveolar bone resorption of four groups was measured by micro-CT and (C) H&E staining. The distances from the CEJ to ABC are marked in black. (D) TRAP staining. Red arrows indicate TRAP-positive surfaces. (E and F) The percentage of bone volume over tissue volume (BV/TV) and the distance from the cementoenamel junction (CEJ) to the alveolar bone crest (ABC) were calculated. Data are shown as mean $\pm$ SD with $n$ = 5 or 6 rats per group. Data between two groups were compared using Student's *t* test. *, $P < 0.05$.

revealed the distance from the cementoenamel junction (CEJ) to the alveolar bone crest (ABC) and the percentage of bone volume over tissue volume (BV/TV), which further confirmed the significant resorption of alveolar bone in the OMVs group (Fig. 2E and F). Morphometric measurements of H&E-stained tissue sections indicated similar

results to micro-CT—the distances from the CEJ to ABC were greater in the OMVs group and CP+OMVs group, respectively (Fig. 2C). Tartrate-resistant acid phosphatase (TRAP)-positive surfaces in the OMVs and CP+OMVs groups were, respectively, increased compared to the control and CP model groups (Fig. 2D).

**P. gingivalis OMVs decreased cell viability in hPDLCs.** The supernatant containing Dil (1,1′-dioctadecyl-3,3,3′,3′-tetramethylindocarbocyanine perchlorate)-labeled *P. gingivalis* OMVs were cocultured with hPDLCs to demonstrate that *P. gingivalis* OMVs were incorporated into hPDLCs (Fig. 3A). Different from *P. gingivalis* lipopolysaccharide (LPS), *P. gingivalis* OMVs significantly decreased the viability of hPDLCs in a time- and concentration-dependent manner (Fig. 3B and Fig. S3). In the LIVE/DEAD viability/cytotoxicity assay, live cells were represented by green fluorescence, while dead cells were represented by red fluorescence in the fluorescence imaging experiments; therefore, propidium iodide (PI) staining also showed that *P. gingivalis* OMVs significantly increased cell death (Fig. 3C). Next, we examined whether *P. gingivalis* OMVs can induce mitochondrial dysfunction in hPDLCs. Generally, the decrease of mitochondrial membrane potential (MMP [$\Delta\Psi$m]) is considered to be one of the indicators of mitochondrial dysfunction and an early feature of apoptotic cells. In this experiment, the changes in MMP were measured by using a fluorescent probe, JC-1. As shown in Fig. 3D, in the *P. gingivalis* OMV-stimulated cells, the red fluorescence decreased and the green fluorescence increased compared to control and LPS-stimulated cells, which suggested OMV-stimulated cells had lower mitochondrial membrane potential. These results suggested that *P. gingivalis* OMVs induced the apoptosis of hPDLCs.

**P. gingivalis OMVs promote apoptosis and inflammation in hPDLCs.** As flow cytometric analysis shows in Fig. 3E and F, the calculated cell death was considerably enhanced in cells treated with *P. gingivalis* OMVs versus the control and LPS groups. Furthermore, we evaluated the protein expression levels of three important mediators of apoptosis, including Bax, Bcl-2, p53, and caspase-3 activity. Western blotting showed that treatment with *P. gingivalis* OMVs resulted in a significant increase in the expression level of p53 and downregulation of antiapoptotic protein Bcl-2 compared to the control and LPS groups (Fig. 3G and H). Consistently, direct targets of p53 and proapoptotic proteins NOXA and PUMA (p53-upregulated modulator of apoptosis) were also increased after stimulation of *P. gingivalis* OMVs (Fig. S4A and B). Caspase-3 activity assay also showed that *P. gingivalis* OMVs resulted in a significant increase in the expression levels of caspase-3 versus the control (Fig. 3I). Levels of IL-1$\beta$, IL-6, and TNF-$\alpha$ were measured by enzyme-linked immunosorbent assay (ELISA) using a commercial kit. Relative to the negative control, the OMVs group had significantly increased levels of IL-1$\beta$ and IL-6 (Fig. 3J). NLRP3 (NOD-, LRR-, and pyrin domain-containing protein 3) is a key component to form and activate the NLRP3 inflammasome, which results in the release of proinflammatory cytokines and promotes pyroptosis (34). Western blotting also showed that treatment of *P. gingivalis* OMVs resulted in a significant increase in the expression levels of NLRP3 (Fig. 3G and H). These results showed that the *P. gingivalis* OMVs promoted apoptosis and inflammation in hPDLCs.

**CBX5 is a functional target of sRNA45033 involved in hPDLC apoptosis.** The analysis of transcriptome sequencing results showed that *P. gingivalis* OMVs were involved in gene regulation, mRNA processing, endocytosis, ubiquitination, and the cell cycle process (Fig. 4A and Fig. S5). Data sets were uploaded to the Gene Expression Omnibus. Combined with previous research (17), Fig. 4B showed small RNAs secreted via *P. gingivalis* OMVs, in which sRNA30540, sRNA16418, sRNA43507, sRNA23392, and sRNA45033 were highly expressed. We predicted the functional targets of these small RNAs with relatively high expression through bioinformatic analysis (https://mrmicrot.imsi.athenarc.gr/?r=mrmicrot/index), and we found six possible differentially expressed targeted genes of these msRNAs (Fig. 4C). Then, targeted genes with high scores and associated with apoptosis were further validated by quantitative reverse transcription-PCR (qRT-PCR), Western blotting, and immunocytochemistry, and we finally screened out gene CBX5 (a downstream target of sRNA45033) and found that it was expressed at low levels in *P. gingivalis* OMV-stimulated hPDLCs and tissues (Fig. 4D to G).

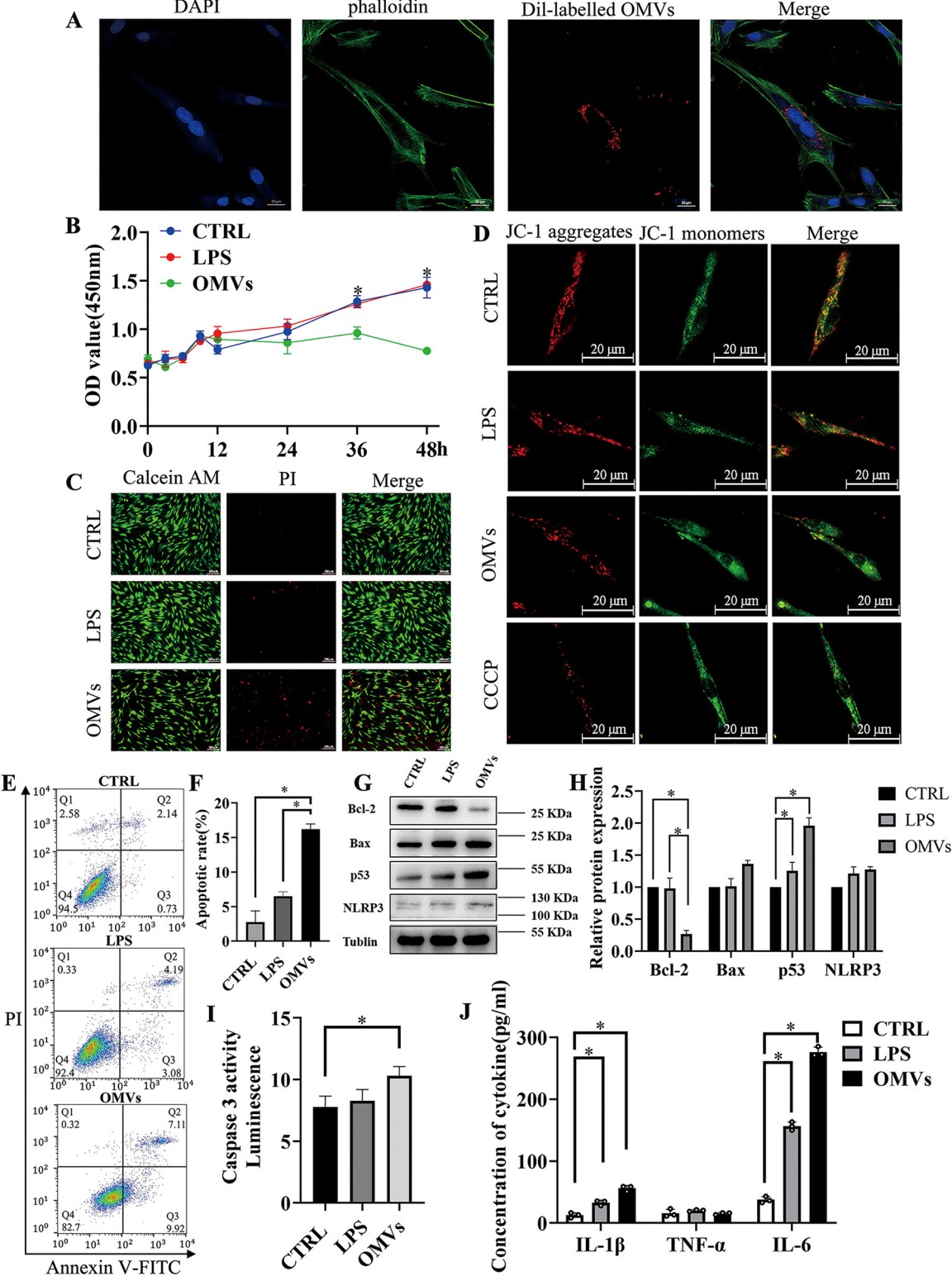

**FIG 3** *P. gingivalis* OMVs decrease cell viability and promote apoptosis and inflammation in hPDLCs. (A) Endocytosis analysis demonstrated that the isolated *P. gingivalis* OMVs were taken up by the hPDLCs. (B) Cell proliferation of hPDLCs stimulated by the LPS or OMVs was measured with the CCK-8 assay. (C) PI staining image of control, LPS-treated, or OMV-treated hPDLCs. (D) The fluorescent probe JC-1 was used to measure mitochondrial membrane potential. CCCP, carbonyl cyanide *m*-chlorophenyl hydrazone. (E and F) Flow cytometry analysis of hPDLCs after being treated with LPS or OMVs. (G and H) Western blotting showed the expression of apoptosis-related proteins like BclII, Bax, p53, and inflammation-associated protein NLRP3 in hPDLCs treated with LPS or OMVs. (I) Caspase 3 activity was measured with the caspase-3 assay kit. (J) Secretion of different cytokines of hPDLCs measured by ELISA. Data are shown as mean ± SD. Data between two groups were compared using Student's *t* test. Cell experiments were conducted three times independently. *, $P < 0.05$.

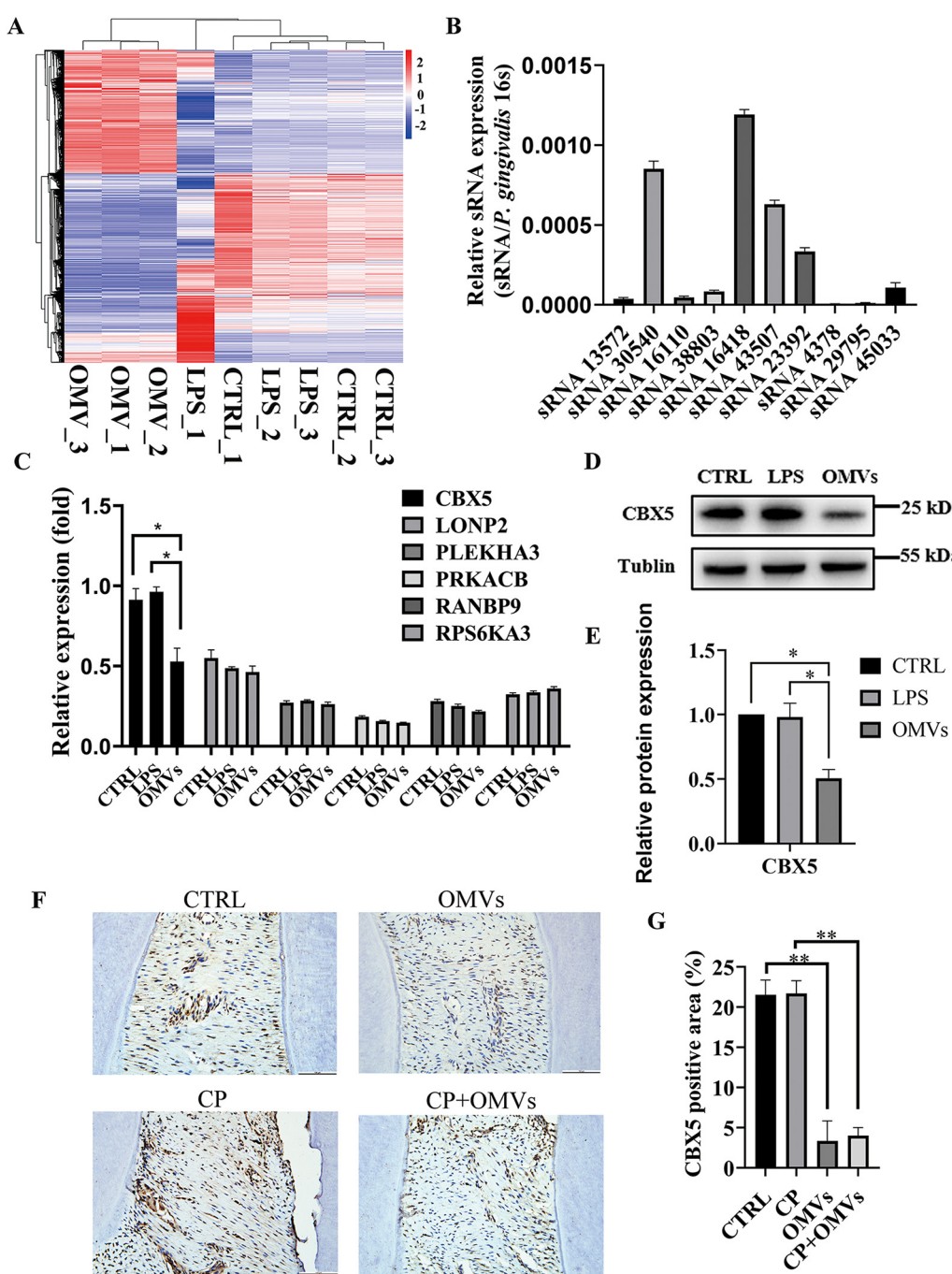

**FIG 4** CBX5 is a functional target of sRNA45033 involved in hPDLC apoptosis. (A) Transcriptome sequencing analysis of hPDLCs administered *P. gingivalis* OMVs; (B) relative expression of msRNA in *P. gingivalis* OMVs; (C) possible target screening of sRNA45033; (D and E) CBX5 expression of hPDLCs treated with LPS or OMVs; (F and G) CBX5 immunochemistry analysis of the CTRL group, CP group, OMVs group, and CP+OMVs group. The positive areas in four independently chosen fields were counted and averaged. Data are shown as mean ± SD. Data between two groups were compared using Student's *t* test. Cell experiments were conducted three times independently. *, $P < 0.05$; **, $P < 0.01$.

**sRNA45033 directly bound to the 3′ UTR of CBX5 and regulate hPDLCs apoptosis.** To ascertain whether sRNA45033 directly binds to the 3′ untranslated region (UTR) of CBX5 and causes translational inhibition, CBX5 wild-type (pCBX5-WT) and mutant (pCBX5-Mut) plasmids were constructed. sRNA45033 mimics (but not the control group) significantly decreased the luciferase activity of the reporter containing the 3′ UTR wild type (WT) of CBX5 compared to the mutant counterpart (Fig. 5A and B). The luciferase assay confirmed that sRNA45033 targeted the 3′ UTR of CBX5 and reduced the expression of CBX5.

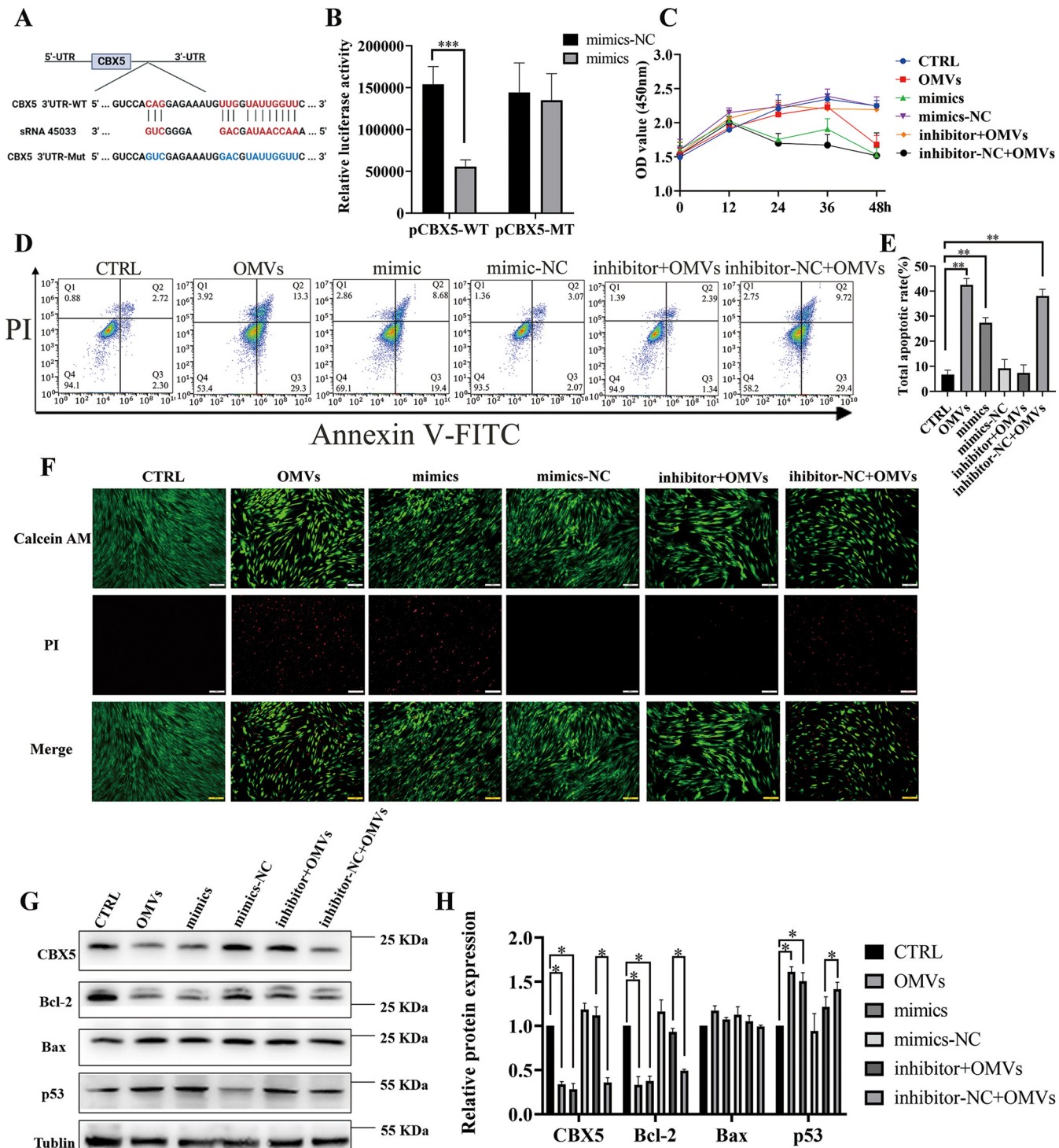

**FIG 5** sRNA45033 directly bound to the 3′ UTR of CBX5 and regulates hPDLC apoptosis. (A) Schematic diagram of mutant strategy of the 3′ UTR of CBX5; (B) dual-luciferase analysis of plasmids with wild-type or mutant 3′ UTR of CBX5; (C) hPDLC proliferation in the OMVs, sRNA45033 mimics, mimics−NC, inhibitor+OMVs, and inhibitor−NC+OMVs groups measured by CCK-8 assay; (D and E) flow cytometry analysis; (F) PI staining; (G and H) Western blot and densitometric analyses of each group. Data are shown as mean ± SD. Data between two groups were compared using Student's *t* test. Cell experiments were conducted three times independently. *, $P < 0.05$; **, $P < 0.01$; ***, $P < 0.001$. NC indicates negative control.

Next, we explored the role of sRNA45033 in cell apoptosis. The Cell Counting Kit-8 (CCK-8) assay results showed that sRNA45033 mimics in hPDLCs caused a significant decrease in the viability of hPDLCs, and inhibition of sRNA45033 dramatically reversed cell viability compared to the results observed in the *P. gingivalis* OMV-stimulated group

(Fig. 5C). PI staining and flow cytometric analysis both showed that sRNA45033 mimics significantly increased cell apoptosis, while inhibition of sRNA45033 significantly reversed cell apoptosis compared to the results observed in the *P. gingivalis* OMV-stimulated group (Fig. 5D to F). sRNA45033 mimic-transfected cells displayed a similar pattern to the *P. gingivalis* OMV-stimulated group, the apoptosis marker p53 showed a significant increase in expression, while Bcl-2 showed a significant downregulation. The sRNA45033 inhibitor group displayed a reverse trend (Fig. 5G and H). PUMA and NOXA were also increased when treated with the sRNA45033 mimic and *P. gingivalis* OMVs and decreased when treated with sRNA45033 inhibitor (Fig. S4C and D). These results supported the hypothesis that sRNA45033 induces cell apoptosis by blocking CBX5 activity.

**CBX5 regulates apoptosis through p53 in *P. gingivalis* OMV-stimulated hPDLCs.** To further understand the regulation of gene *CBX5* on apoptosis, hPDLCs were transfected with an plasmid overexpressing CBX5 (CBX5-OE) and a control vector, a CBX5 knockdown (CBX5-sh) plasmid, and a scramble knockdown plasmid (SC). The overexpressed and knockdown efficiency of CBX5 in hPDLCs was validated by qRT-PCR and Western blotting (Fig. 6A and E to H). CCK-8 showed that cell viability was reduced in the CBX5-sh group compared with those in the control and SC groups and was enhanced in the CBX5-OE+OMVs group compared with that in the OMVs group after transfections in hPDLCs. Furthermore, downregulated CBX5 significantly decreased cell viability compared to those in the control and SC groups, while upregulated CBX5 reversed the inhibitory influence of *P. gingivalis* OMVs (Fig. 6B). Additionally, downregulated CBX5 significantly increased cell apoptosis, consistent with the OMVs group, and overexpressed CBX5 rescued the promotional effect of *P. gingivalis* OMVs on cell apoptosis in hPDLCs, as shown by PI staining and flow cytometric analysis (Fig. 6C, D, and I). More importantly, in the CBX5-sh plasmid-transfected cells, p53 showed a significant increase in expression, while Bcl-2 showed a significant downregulation, displaying a similar pattern to the *P. gingivalis* OMV-stimulated group. While overexpressed CBX5 rescued the increased expression of p53, Bcl-2 was increased (Fig. 6E to H). As expected, PUMA and NOXA showed similar changes in p53 expression (Fig. S4E to H).

**p53 methylation suppressed by *P. gingivalis* OMVs in hPDLCs via CBX5.** To further understand the mechanism of how OMVs control p53 expression, we hypothesized that CBX5 regulates the epigenetic remodeling (H3K9me3, in this case) of the p53 gene, which results in the change in expression. Western blotting confirmed that decreased CBX5 reduced the level of H3K9me3 in hPDLCs (Fig. 7A and B). CUT&Tag (cleavage under targets and tagmentation) analysis was conducted to elucidate the connection between CBX5 and p53. Following the selection of hPDLCs with *P. gingivalis* OMVs, peak calling and transcriptional start site (TSS) profile analysis showed that CBX5 bound to the p53 gene similarly (Fig. 7C and D). For hPDLCs treated with *P. gingivalis* OMVs, the signal intensity was clearly lower than in those treated without OMVs (Fig. 7E). Consistently, the signal intensity of H3K9me3 of the p53 gene treated with *P. gingivalis* OMVs was lower than in those treated without OMVs (Fig. 7F). These results suggested that *P. gingivalis* OMVs reduced the level of CBX5 to regulate the methylation of p53, which determines the fate of hPDLCs (Fig. 8).

## DISCUSSION

The Gram-negative anaerobic bacterium *P. gingivalis* has been considered the keystone pathogen in periodontitis (6). Recent findings have suggested *P. gingivalis* plays an essential role in systemic diseases, such as cardiovascular diseases, rheumatoid arthritis, diabetes, pregnancy problems, Alzheimer's disease, and insulin resistance (35–38). Henceforth, it is imperative to form a deeper understanding of the mechanism of *P. gingivalis* in periodontitis.

OMVs produced by periodontal bacteria have been demonstrated to take part in the progress of periodontal inflammation and tissue destruction (39). Periodontal pathogen OMVs are also related to distant organ dysfunction, including cardiovascular, brain, and joint dysfunction (40–42). Despite the explored effects in the periodontium, in-depth knowledge of the roles of these vesicles would significantly contribute to understanding the development of periodontitis and its related systemic diseases (43, 44).

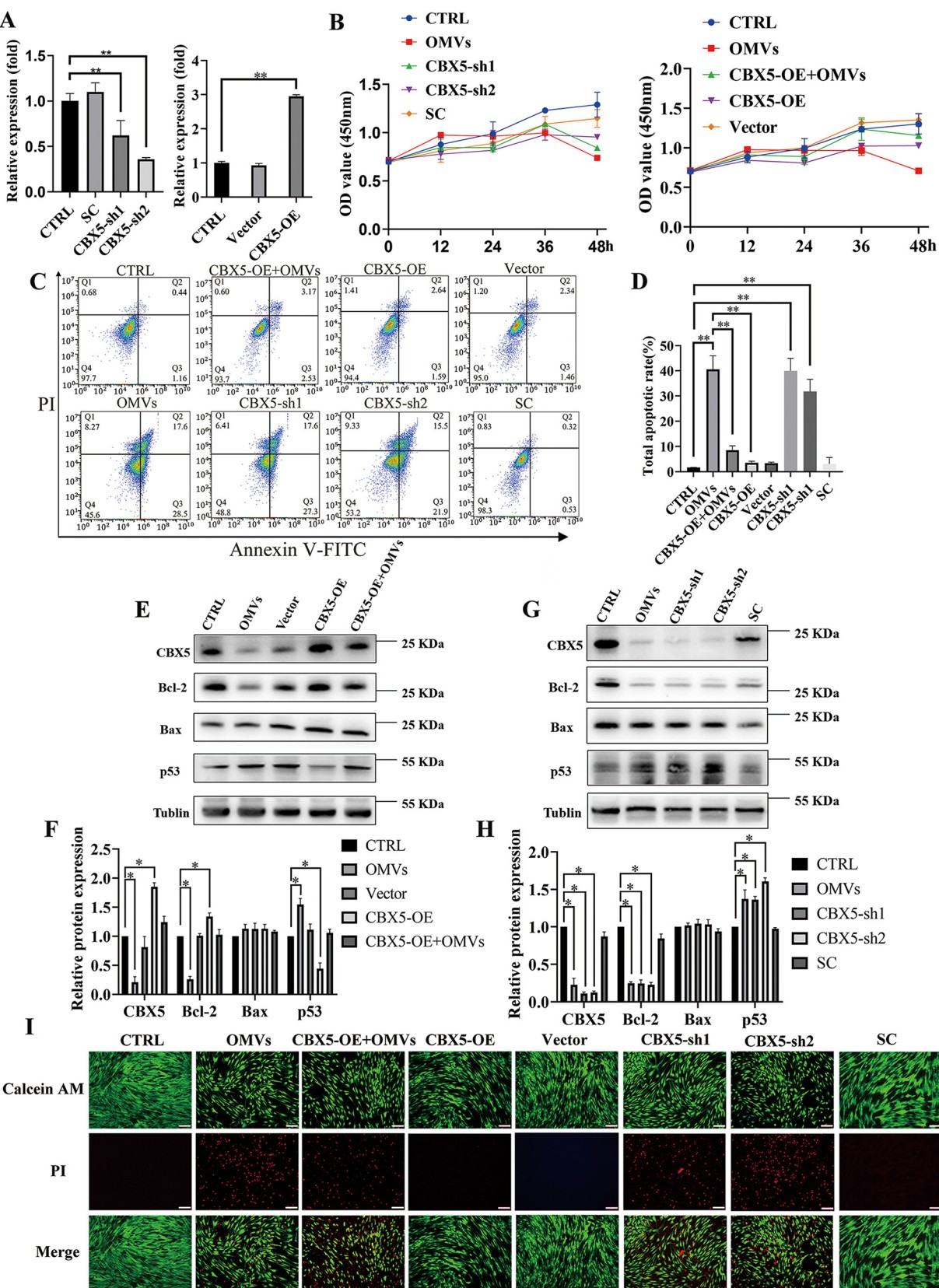

**FIG 6** CBX5 regulates apoptosis via p53/Bcl-2 axis. (A) The efficacy of knockdown and overexpression of CBX5 was measured by qRT-PCR. (B) CCK-8 analysis of CBX5 knockdown and overexpression in hPDLCs; (C and D) flow cytometry analysis; (E to H) Western blot and the densitometric analyses of each group; (I) PI staining after overexpression and knockdown of CBX5. Data are shown as mean ± SD. Data between two groups were compared using Student's *t* test. Cell experiments were conducted three times independently. *, $P < 0.05$; **, $P < 0.01$.

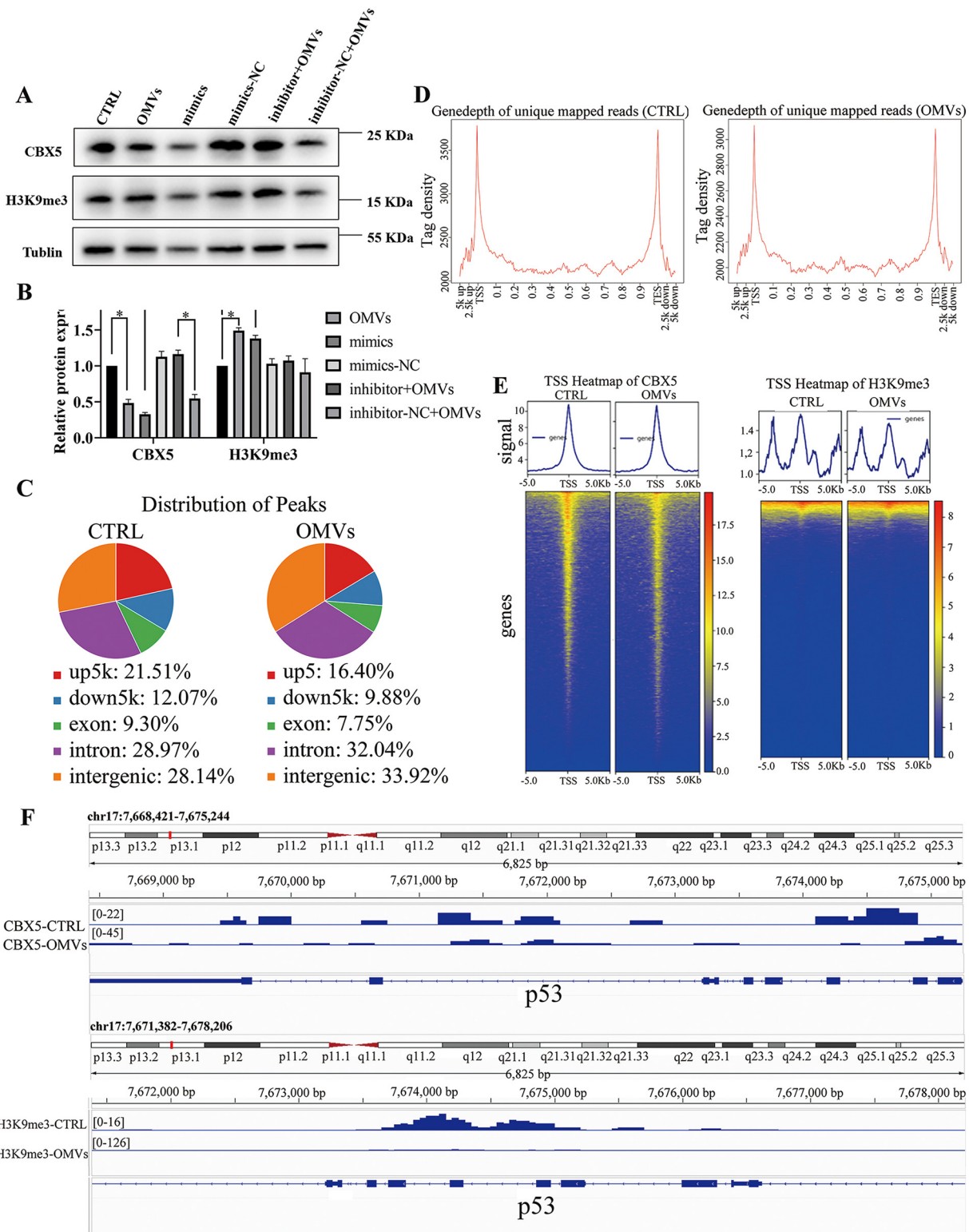

**FIG 7** CBX5 regulates apoptosis through methylation in *P. gingivalis* OMV-stimulated hPDLCs. (A and B) The methylation level was measured by Western blotting. (C and D) The CUT&Tag technique was employed. Peak calling analysis of hPDLCs treated with or without *P. gingivalis* OMVs was performed. (E) TSS heat map of CBX5 and H3K9me3; (F) Iintegrative Genomics Viewer showing the signals of CBX5 and H3K9me3 at p53 loci. *, $P < 0.05$.

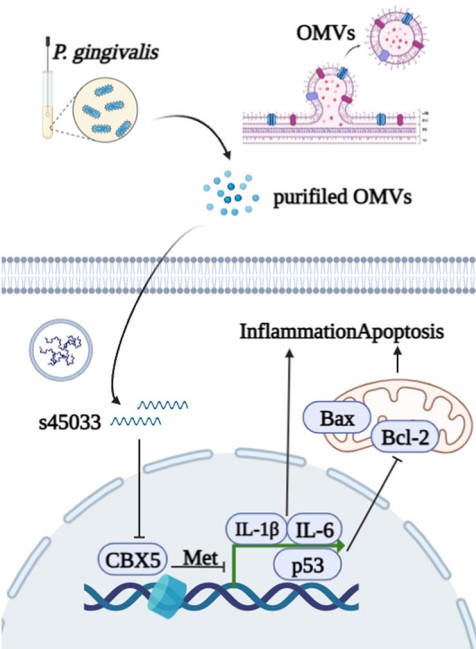

**FIG 8** Schematic diagram of sRNA45033 packaged by *P. gingivalis* OMVs suppressing CBX5 to regulate apoptosis through p53 in hPDLCs.

Previous research had found that *P. gingivalis* was the most prolific OMV producer among the most proportionally increased bacteria during periodontal disease (45). *P. gingivalis* OMVs are actively involved in the progress of periodontal inflammation, immunomodulation of the host, and tissue destruction (46, 47). The average diameter of OMVs is 202.8 $\pm$ 71.87 nm, which was consistent with the range of the former study (11, 48, 49). The different average diameters could be result from OMVs' fusion after ultracentrifugation or osmotic pressure after elution. The size difference is worth exploring because there is possibility that the size of OMVs could determine the biofunction of OMVs (50). The nanosized *P. gingivalis* OMVs could invade cells, as we confirmed in the endocytosis assay that hPDLCs successfully took up isolated *P. gingivalis* OMVs. Both silver staining and Coomassie blue staining showed that the protein bands were similar between *P. gingivalis* and *P. gingvalis* OMVs. However, the different distributions of protein content suggested their different functions in host-pathogen interaction. The isolated *P. gingivalis* OMVs were confirmed to promote bone resorption *in vivo*. Transcriptome sequencing (RNA-seq) was applied to compare the different functions between *P. gingivalis* OMVs and LPS. The annotated pathways of *P. gingivalis* OMVs were related to gene regulation, mRNA processing, and cell cycle process, while LPS was involved in inflammatory pathways. To investigate the detailed biofunction of isolated *P. gingivalis* OMVs, OMV-administered hPDLCs were found to be less proliferative and reduced viability. Generally, the decrease of mitochondrial membrane potential is considered to be one of the indicators of mitochondrial dysfunction and an early feature of apoptotic cells. The irregulated MMP also indicated aberrant reactive oxygen species (ROS) levels and showed signs of early apoptosis stimulated by *P. gingivalis* OMVs. NLRP3 and the proinflammatory cytokines IL-1$\beta$ and IL-6 were much more elevated, which was consistent with other studies (51). TNF-$\alpha$ was also evaluated, however, with few changes: this was probably caused by the complex content within the *P. gingivalis* OMVs. Choi and colleagues found that only IL-5, IL-13, and IL-15 were changed in Jurkat T cells after transfections with synthetic msRNAs (17). Thus, future study of the relationship between *P. gingivalis* OMVs and proinflammatory cytokines will also help to clarify the roles of *P. gingivalis* OMVs in the pathogenesis of periodontitis. The apoptosis level was significantly higher when hPDLCs were treated with *P. gingivalis* OMVs, and the expression changes of apoptosis-related proteins p53, Bcl-2, and Bax indicated a connection between *P. gingivalis* OMVs and the apoptosis pathway.

OMVs contain genetic material that has been suggested as a novel gene transfer system (16). We hypothesized that msRNAs within the *P. gingivalis* OMVs regulate host genes in a way like miRNA, which binds to target mRNA. We selected the 10 highest predicted msRNAs from former sequencing results to identify the msRNA regulating apoptosis in hPDLCs. Five more highly expressed msRNAs in the isolated *P. gingivalis* OMVs were then verified and selected to analyze their downstream targets, and CBX5 was screened out as a possible candidate.

CBX5 is an important architectural protein that forms heterochromatin, which is compact in structure and transcriptionally repressed. CBX5 induces epigenetic remodeling like H3K9 methylation in the promoter region and silences the adjacent gene (52). The expression level of CBX5 was confirmed to be significantly reduced after OMV administration. To understand the mechanism of this regulation, the binding of sRNA45033 and CBX5 3′ UTR was confirmed by a dual-luciferase reporter assay. The sRNA45033 mimic and inhibitor confirmed that sRNA45033 inhibited cell proliferation and regulated apoptosis via CBX5 expression.

p53 induces apoptosis through upregulation of proapoptotic BH3-only members of the Bcl-2 protein family (such as BIM, NOXA, and PUMA), which bind and inhibit prosurvival Bcl-2 proteins, in turn, releasing proapoptotic Bcl-2 family members like Bax (53). Studies have suggested that CBX5 was (i) involved in inflammatory response-related apoptosis genes, (ii) repressed apoptosis in renal cell carcinoma, (iii) interacted with p53 in several cancer cell lines, and (iv) and was expressed to a higher level with p53 loss in some cell lines (31, 54–56). These prompted us to investigate the underlying mechanism of CBX5 regulating apoptosis. Indeed, overexpression and knockdown of CBX5 further proved that CBX5 is a key regulator between sRNA45033 and p53/Bcl-2/Bax.

CUT&Tag is a novel technique for mapping chromatin features and understanding epigenetic regulation in the genome. In short, a specific antibody binds to a chromatin protein and then tethers with a protein A-Tn*5* transposase fusion protein. Upon activation, this protein generates sequencing libraries with remarkable signal-to-noise ratio and reliability (57, 58). To explore the underlying mechanism, CUT&Tag analysis and Western blotting were carried out, and we found that CBX5 controlled the H3K9me3 level of p53, which forms heterochromatin to affect p53 expression. Subsequently, Bcl-2 was downregulated by the elevated p53 expression and the host cells underwent apoptosis.

This study uncovered a novel complex regulatory system of *P. gingivalis* OMVs; however, there are some limitations to this study. First, *P. gingivalis* OMVs contain many other msRNAs that also possibly contribute to the progress of periodontitis. Second, we verified only the p53/Bcl-2/Bax axis, yet apoptosis regulation is an extremely complicated network. The expression of Bax was only slightly changed after the administration of OMVs; this might be caused by the internal high level of Bax in hPDLCs. Third, to further confirm the methylation of the p53 and involvement of CBX5, chromatin immunoprecipitation will be utilized in the future. Fourth, more detailed *in vivo* experiments are needed to provide solid evidence on the roles of *P. gingivalis* OMVs and identify their effectiveness as possible treatment targets. Due to these limitations, we will further explore the role and function of *P. gingivalis* OMVs systemically. A more in-depth understanding of *P. gingivalis* OMVs periodontitis will also be beneficial in the aspect of treating periodontitis.

In summary, our data have suggested a novel mechanism in which sRNA45033 packaged by *P. gingivalis* OMVs inhibits CBX5 expression in host cells, which lowers the H3K9me3 level of p53. Accordingly, p53 expression was elevated and Bcl-2, the key protein in apoptosis, was downregulated and ultimately promoted apoptosis in the host cells.

## MATERIALS AND METHODS

**Bacterial cultures.** *P. gingivalis* ATCC 33277 was purchased from the Shanghai Biology Collection Center (Shanghai, China), and the culture method was adopted from previously studies (17, 59). Briefly, cells were cultured in brain heart infusion broth (Oxoid, Basingstoke, United Kingdom) containing 10% sterile defibrinated sheep blood, 0.5% hemin (Hopebiol, Shandong, China), and 0.1% vitamin $K_1$ (Hopebiol, Shandong, China) in an anaerobic chamber (80% $N_2$, 10% $O_2$, 10% $H_2$) at 37°C. Bacteria were harvested after growing for 48 h under anaerobic conditions in an anaerobic chamber.

**Isolation and identification of *P. gingivalis* OMVs.** The isolation of *P. gingivalis* OMVs followed established protocols (47, 60). Briefly, for freshly grown *P. gingivalis* cells (optical density at 600 nm [$OD_{600}$] of 1.0,

equivalent to $9 \times 10^9$ CFU), the bacterial culture medium was centrifuged at $8,000 \times g$ for 15 min at 4℃ to remove the bacterial cells. The supernatant was collected and filtered using a 0.22-$\mu$m-pore syringe filter (Millipore, MA, USA). The samples were centrifuged at $100,000 \times g$ for 2 h at 4℃ (Fig. 1A). The OMV fractions were dissolved in 200 $\mu$L of phosphate-buffered saline (PBS) and stored at $-80$℃ until needed.

OMVs were analyzed by transmission electron microscopy (TEM). The OMV samples were applied to 200-mesh Formvar-carbon grids (Ted Pella) with 2% phosphotungstic acid (Solarbio, Beijing, China). Samples were dried on the grids, viewed, and photographed with a Hitachi model H-7500 transmission electron microscope (Hitachi, Tokyo, Japan). The diameter of *P. gingivalis* OMVs was measured using dynamic light scattering (Malvern Zetasizer Nano ZS90; Malvern, United Kingdom).

For silver staining, concentrations of protein were measured using a bicinchoninic acid (BCA) protein assay kit (Beyotime, Shanghai, China). *P. gingivalis* OMVs (400 ng of total protein) and an equal amount of *P. gingivalis* proteins were separated by SDS-PAGE. The proteins on SDS gels were visualized by using Silver Stain Plus (Beyotime Biotechnology, Nanjing, China).

**Animals.** The animal study was reviewed and approved by Institutional Animal Care and Use Committee of Nanjing Medical University (IACUC-2006020). SD rats (male, 6 to 7 weeks old) were purchased from the Animal Core Facility of Nanjing Medical University. They were caged under specific-pathogen-free (SPF) conditions, fed a normal diet, and subjected to constant temperature and humidity for 12 h each of alternating light/darkness cycles. SD rats were randomly divided into four groups: (i) the control rats (CTRL group); (ii) healthy rats treated with *P. gingivalis* OMVs (OMVs group); (iii) the chronic periodontitis model rats (CP group), and (iv) chronic periodontitis rats treated with *P. gingivalis* OMVs (CP+OMVs group). The chronic periodontitis modeling method was conducted as follows. A standard ligation wire (0.25 mm) was passed through the proximal and distal adjacent spaces of the bilateral maxillary M1, ligated around the dental neck, placed in the gingival sulcus as far as possible, and knotted in the mesiobuccal side of M1 (45). The ligation wire was checked daily for shedding and was religated when the wire detached. Periodontal ligation was maintained for 4 weeks to keep periodontal tissue inflammation in a stable state to establish the model of periodontitis (46, 47). Two groups received *P. gingivalis* OMVs at a dose (20 $\mu$L) of 5 $\mu$g/$\mu$L three times per week by local injection. The other group of rats received a comparable volume of PBS (Fig. 2A). In the animal experiments, the investigator was blind to the group allocation. After 4 weeks, the rats were euthanized and the femurs were harvested for microcomputed tomography (micro-CT) and histological analysis.

**Micro-CT and histological analysis.** The rats were sacrificed 4 weeks after surgery, and the cranial tissue was collected. The tissues were fixed with 4% paraformaldehyde (PFA) for 1 week and 75% ethanol for micro-CT evaluation. 3D images of the mineralized tissues were reconstructed using Sky scan software. The bone volume/tissue volume (BV/TV) of each sample was collected for analysis. After micro-CT evaluation, the tissues were demineralized in 14% EDTA solution for 3 months, dehydrated by an automatic dehydrator, and embedded in paraffin. Paraffin sections were cut into tissue sections 5 mm thick for hematoxylin and eosin (H&E), TRAP staining, and immunochemistry.

**Cell culture.** The cell study was reviewed and approved by Ethics Committee of the Affiliated Stomatological Hospital of Nanjing Medical University (PJ2021-089-001). hPDLCs were chosen to conduct *in vitro* studies (61). hPDLCs were extracted from the middle third of the root surfaces of young patients (12 to 16 years old) requiring orthodontic treatment, and informed consent was obtained. The obtained hPDLCs were confirmed by the immunochemistry staining of keratin and vimentin (see Fig. S2 in the supplemental material). The cells were then cultured in $\alpha$ minimum essential medium ($\alpha$-MEM) supplemented with 15% fetal bovine serum (FBS) (Gibco, CA, USA) and 1% streptomycin-penicillin (Gibco, CA, USA) and incubated at 37℃ in a humidified atmosphere with 5% $CO_2$. Upon attaining 90% confluence, hPDLCs were maintained in $\alpha$-MEM with 10% FBS and used between passages 3 and 6.

**Endocytosis analysis.** DiI (1,1′-dioctadecyl-3,3,3′,3′-tetramethylindocarbocyanine perchlorate) was used to label the *P. gingivalis* OMVs' lipid membrane. The isolated *P. gingivalis* OMVs were incubated with 2 $\mu$L of DiI (Molecular Probes, CA, USA) in 100 $\mu$L PBS for 15 min. Subsequently, the DiI-labeled *P. gingivalis* OMVs were harvested and incubated with hPDLCs at 37℃ for 6 h. Then, the cells were fixed with 4% paraformaldehyde for 20 min and stained with phalloidin and DAPI (4′,6-diamidino-2-phenylindole) (Beyotime Biotechnology, Shanghai, China). Fluorescence images were captured under a confocal microscope (Leica, Heidelberg, Germany).

**Cell Counting Kit-8 analysis.** To investigate the effects of *P. gingivalis* OMVs (10 $\mu$g/mL) on the growth of cultured cells, a medium containing $1 \times 10^5$ hPDLCs was pipetted into 96-well tissue culture plates. The cells were allowed to attach and grow for 48 h. After the indicated treatments, the cells were washed with PBS buffer three times, followed by the addition of 100 $\mu$L $\alpha$-MEM medium supplemented with 10% Cell Counting Kit-8 (CCK-8), and then the cells were incubated at 37℃ for 2 h. The optical density at 450 nm ($OD_{450}$) was measured by a microplate reader (SpectraMax190; Molecular Devices, San Jose, CA, USA). At least three separate experiments were done for statistical analysis.

**Calcein/propidium iodide staining.** Medium containing $2 \times 10^4$ hPDLCs was pipetted into 24-well plates, and the cells were allowed to attach and grow for 48 h. After the indicated treatments, the cells were washed with PBS buffer three times and then were analyzed with the LIVE/DEAD viability/cytotoxicity assay kit for animal cells (KeyGEN BioTECH, Jiangsu, China), with Calcein AM and PI for 30 min, which could stain the cells to distinguish the living cells (green) from the dead ones (red). The samples were then observed under a fluorescence microscope (DMI6000B; Leica, Heidelberg, Germany).

**JC-1 staining for mitochondrial membrane potential.** Mitochondrial membrane potential was measured using the mitochondrial membrane potential probe JC-1 staining dye in hPDLCs. Briefly, cells were cultured in a glass bottom dish with or without pretreatment with *P. gingivalis* OMVs (10 $\mu$g/mL) for 24 h. After JC-1 working solution was added, the cells were maintained in a $CO_2$ incubator for 20 min.

**TABLE 1** Primers for msRNA screening

| Primer | Sequence |
| --- | --- |
| sRNA45033-RT | GTCGTATCCAGTGCGTGTCGTGGAGTCGGCAATTGCACTGGATACGACCAGCCCT |
| sRNA45033-F | AGCGAGGGAAACCAATAGCAG |
| sRNA38803-RT | GTCGTATCCAGTGCGTGTCGTGGAGTCGGCAATTGCACTGGATACGACGGCCGTA |
| sRNA38803-F | AGGTAAGGGACAGGGACAGA |
| sRNA4378-RT | GTCGTATCCAGTGCGTGTCGTGGAGTCGGCAATTGCACTGGATACGACTAACCCA |
| sRNA4378-F | TCTGAGGGTGGGATTATGAGCTA |
| sRNA29795-RT | GTCGTATCCAGTGCGTGTCGTGGAGTCGGCAATTGCACTGGATACGACGGGTTC |
| sRNA29795-F | GCTGAGGGTGGGAAATGAAGT |
| sRNA16418-RT | GTCGTATCCAGTGCGTGTCGTGGAGTCGGCAATTGCACTGGATACGACCTGGCAA |
| sRNA16418-F | GTGGGTTACACCGGACCTC |
| sRNA43507-RT | GTCGTATCCAGTGCGTGTCGTGGAGTCGGCAATTGCACTGGATACGACGCCAGAG |
| sRNA43507-F | TGGGATCAGTTGGTTGGAAAGA |
| sRNA23392-RT | GTCGTATCCAGTGCGTGTCGTGGAGTCGGCAATTGCACTGGATACGACTGTCACC |
| sRNA23392-F | CTGGTGGGATAAAGCGAGAGG |
| sRNA30540-RT | GTCGTATCCAGTGCGTGTCGTGGAGTCGGCAATTGCACTGGATACGACGCTGACA |
| sRNA30540-F | GGGTTGCAGGACGCGAT |
| sRNA13572-RT | GTCGTATCCAGTGCGTGTCGTGGAGTCGGCAATTGCACTGGATACGACGCAGAG |
| sRNA13572-F | GCGAGGTGGGATAAGGAAGC |
| sRNA16110-RT | GTCGTATCCAGTGCGTGTCGTGGAGTCGGCAATTGCACTGGATACGACGTAGCAG |
| sRNA16110-F | GCGTGGCATTGGTATTGTTGG |
| *P. gingivalis* 16S RNA-F | TGTAGATGACTGATGGTGAAAACC |
| *P. gingivalis* 16S RNA-R | ACGTCATCCACACCTTCCTC |
| sRNA-R | CAGTGCGTGTCGTGGAGT |
| GAPDH-F | CTGGGCTACACTGAGCACC |
| GAPDH-R | AAGTGGTCGTTGAGGGCAATG |
| β-Actin-F | GTCCCTCACCCTCCCAAAAG |
| β-Actin-R | GCTGCCTCAACACCTCAACCC |

The staining solution was removed, and then the cells were gently washed twice with JC-1 staining buffer. Fluorescence was detected by confocal microscopy (Leica, Heidelberg, Germany).

**Quantitative reverse transcription-PCR analysis of sRNAs and mRNAs.** Total RNA was isolated by the RNA Simple Total kit (TianGen, Beijing, China), and *P. gingivalis* OMV sRNAs were isolated by the RNAeasy small RNA isolation kit (Beyotime Biotechnology, Shanghai, China). In addition, cDNA was generated with a PrimeScript RT master mix for quantitative PCR (qPCR) (TaKaRa, Kyoto, Japan), followed by analysis using an SYBR Premix *Ex Taq* II (TaKaRa, Kyoto, Japan). Stem-loop reverse transcription was utilized to reverse msRNAs because of the short fragment. PCR conditions consisted of an initial 30 s of denaturation at 95°C, followed by 40 cycles of 95°C for 5 s, and 60°C for 30 s. All reactions were performed in triplicate. Change in transcript abundance of all tested genes was calculated using the threshold cycle ($2^{-\Delta\Delta CT}$) method. The primers used in the study are listed in Table 1 and Table 2.

**RNA library construction and sequencing.** A total amount of 1 $\mu$g RNA per sample was used as input material for the RNA sample preparations. Sequencing libraries were generated using NEBNext Ultra RNA library prep kit for Illumina (NEB, MA, USA) following the manufacturer's recommendations, and index codes were added to attribute sequences to each sample. Briefly, mRNA was purified from total RNA using poly(T) oligonucleotide-attached magnetic beads. Fragmentation was carried out using divalent cations under elevated temperature in NEBNext first-strand synthesis reaction buffer (5×). First-strand cDNA was synthesized using random hexamer primer and Moloney murine leukemia virus (MMuLV) reverse transcriptase (RNase H$^-$). Second-strand cDNA synthesis was subsequently performed using DNA polymerase I and RNase H. The remaining overhangs were converted into blunt ends via exonuclease/polymerase activities. After adenylation of 3′ ends of DNA fragments, NEBNext adaptor with a hairpin loop structure was ligated to prepare for hybridization. To select cDNA fragments preferentially 250 to ~300 bp in length, the library fragments were purified with the AMPure XP system (Beckman Coulter, CA, USA). Then, 3 $\mu$L USER enzyme (NEB, MA, USA) was used with size-selected, adaptor-ligated cDNA at 37°C for 15 min, followed by 5 min at 95°C before PCR. Then, PCR was performed with Phusion high-fidelity DNA polymerase, universal PCR primers, and index (X) primer. Finally, PCR products were purified (AMPure XP system) and library quality was assessed on the Agilent Bioanalyzer 2100 system.

The clustering of the index-coded samples was performed on a cBot cluster generation system using TruSeq PE cluster kit v.3-cBot-HS (Illumia, NEB, MA, USA) according to the manufacturer's instructions. After cluster generation, the library preparations were sequenced on an Illumina Novaseq platform and 150-bp paired-end reads were generated.

**Western blot.** After being washed with PBS, hPDLCs were lysed in lysis buffer (whole-cell lysis assay; KeyGEN BioTECH, Jiangsu, China). Concentrations of protein were measured using a BCA protein assay kit. Proteins were separated on SDS-polyacrylamide gels and transferred to polyvinylidene difluoride (PVDF) membranes (Millipore, Bedford, MA, USA). Membranes were incubated with primary antibodies to Bcl-2 (1:1,000) (Abcam, Cambridge, United Kingdom), Bax (1:1,000) (Abcam, Cambridge, United

**TABLE 2** Primers for qRT-PCR

| Primer | Gene | Sequence Forward | Reverse |
|---|---|---|---|
| s43507 | ENST00000321801(BOLL) | TGCCTTTGAATAACCCAACAAGT | TTCACAGACCCATACTGGGAA |
| | ENST00000611864(MGAT4C) | TCACCTATCGCTACCTAGCTG | GGCATCACGCCAGGAAGAAT |
| | ENST00000261427(UBE2K) | GTTCCGTCACAGGGGCTATTT | AATACCGTGCGGAGAGTCATT |
| | ENST00000321662(GPR137C) | ACCTGGCGGAGGTTATATGTAA | TCTCCATGAACTAGCATTGCG |
| | ENST00000410080(PRPF40A) | AGAGAGCGAATATGCCTCCTG | GCATTACTGACGACATCATTCCA |
| | | | |
| s16418 | ENST00000303177(HMP19) | GCCGCCTTCAGTTGAGGAT | TCCGGCTGATATTCCGTTCTT |
| | ENST00000244745(SOX4) | GCCGAGTGGAAACTTTTGTCG | GGCAGCGTGTACTTATCCTTCT |
| | ENST00000260197(SORL1) | CAAGGTGTACGGACAGGTTAGT | CCAATGCCAGGCTATCTCG |
| | ENST00000370598(ADGRB3) | TGCCCAAGACTTCTGGTGTTC | AAGTTAGAGCAGCTAAGGTCCT |
| | ENST00000261798(CSNK1A1) | AGTGGCAGTGAAGCTAGAATCT | CGCCCAATACCCATTAGGAAGTT |
| | | | |
| s30540 | ENST00000615466(ZNF189) | AGCCTTTCGATTAAGCACATACC | AGCTCCGACTGAAACTTTTTCC |
| | ENST00000266643(MARCH9) | ATCTCCCTGACGGTCATCGAG | GGCTGAAGGGCTGAGTGAG |
| | ENST00000370689(PRKACB) | CCATGCACGGTTCTATGCAG | GTCTGTGACCTGGATATAGCCTT |
| | ENST00000320848(MRFAP1L1) | CCCCTGGACATAGACGAGGT | CCGTGCTCGCGTATAAGAGAC |
| | ENST00000356450(NUDT10) | CCAGACACGGACCTACGA | CCTTGACTCCCGCCTCTT |
| | | | |
| s23392 | ENST00000379565(RPS6KA3) | CGCTGAGAATGGACAGCAAAT | TCCAAATGATCCCTGCCCTAAT |
| | ENST00000614987(RPS6KA5) | CTCCTCACTGTCAAGCACGAG | GCCTTTTGAACGATTGTTGCCT |
| | ENST00000011619(RANBP9) | GGCCGTGGACGAACAAGAG | GAACTGACGCGGCATCTTTT |
| | ENST00000234453(PLEKHA3) | ACTGTGACCTCTTAATGCAGC | CTCAAGCGTTGTGATGAATGTG |
| | ENST00000491143(ONECUT2) | GGAATCCAAAACCGTGGAGTAA | CTCTTTGCGTTTGCACGCTG |
| | | | |
| s38803 | ENST00000542274(CDH3) | ATCATCGTGACCGACCAGAAT | GACTCCCTCTAAGACACTCCC |
| | ENST00000614512(SATB2) | CATGCCACAGTCCGCAATG | GGCCCAGAACACAATAGTCTGA |
| | ENST00000338965(NCR3LG1) | TGTGAGTCAAGTGGGTTCTACC | CATGCCGTACCACACACTG |
| | ENST00000285737(LONP2) | ATGCTGTGAGCCTAGAGGAG | GCTCTGGCATACTAGATGTTCG |
| | ENST00000267890(TTBK2) | CAATCAACGCACATCGGAACA | GAGCCTACTTGCTCCTTGTCC |
| | | | |
| s45033 | ENST00000209875(CBX5) | AACAGTGCCGATGACATCAAA | GCCCCAATGATCTTTTCTGGT |
| | ENST00000335420(KLHDC10) | TACGATGGGACCCAGTTAGGA | TGTGGCCTCTCAAAAACCTGT |
| | ENST00000358127(PAX5) | AAACCAAAGGTCGCCACAC | GTTGATGGAACTGACGCTAGG |
| | ENST00000333137(SMTN) | GGGATCGTGTCCACAAGTTCA | GCTACTCCTCGTTGCTCCTT |
| | ENST00000548729(POC1B-GALNT4) | AATACTATGCCTCCCTTTG | AGCCCACTTTCAGTTTCA |

Kingdom), p53 (1:1,000) (Proteintech, Chicago, IL, USA), PUMA (1:1,000) (Proteintech, Chicago, IL, USA), NOXA (1:1,000) (Affinity Biosciences, Cincinnati, OH, USA), CBX5 (1:500) (Proteintech, Chicago, IL, USA), NLRP3 (1:1,000) (Abcam, Cambridge, United Kingdom), H3K9me3 (1:1,000) (Abcam, Cambridge, United Kingdom), and tubulin (1:1,000) (Proteintech, Chicago, IL, USA) overnight at 4°C. Subsequently, membranes were then incubated with corresponding secondary antibodies (Proteintech, Chicago, IL, USA) at room temperature for 60 min. Blotted bands were visualized after ECL enhanced chemiluminescence exposure by a chemiluminescent gel imaging system (Tanon, Shanghai, China). The protein levels were analyzed using ImageJ software; at least three biological replicates were included.

**Flow cytometry.** We analyzed the apoptosis of hPDLCs using flow cytometry. After the indicated treatments, for the cell apoptosis analysis, hPDLCs were collected separately and incubated with the Alexa Fluor 488-annexin V/dead cell apoptosis kit (Invitrogen, CA, USA), using annexin V-fluorescein isothiocyanate-propidium iodide, and the cells were incubated at room temperature for 15 min. Subsequently, the apoptotic cells were analyzed using a FACSCalibur (Becton Dickenson, Franklin Lakes, NJ, USA).

**ELISA.** Il-1$\beta$, TNF-$\alpha$, and IL-6 in the cell supernatant were determined via an enzyme-linked immunosorbent assay (ELISA) by using commercially available ELISA sets (Neobiscience, Shenzhen, China) following the instructions of the manufacturer. All samples were measured in duplicate.

**Caspase-3 activity assay.** Activities of caspase-3 were measured using GreenNuc caspase-3 assay kit for live cells (Beyotime, Shanghai, China). In brief, cells were cultured in 96-well black plates with or without pretreatment with *P. gingivalis* OMVs (10 $\mu$g/mL) for 24 h, and the untreated cells were used as the negative control. The inhibitor group added a certain concentration of inhibitor (5 $\mu$M). The cells were replaced with fresh culture medium containing substrate (5 $\mu$M) or PBS. The Ac-DEVD-CHO inhibitor group was supplemented with the original concentration of inhibitor (5 $\mu$M). Cells were incubated at room temperature in dark for 15 to 30 min (longer if necessary). The microplate reader (SpectraMax 190; Molecular Devices) was set with an excitation wavelength of 485 nm and an emission wavelength of 515 nm. For adherent cells, bottom reading was recommended.

**Dual-luciferase reporter assay.** pEZX-FR02 luciferase reporter vector (GeneCopoeia, MD, USA) containing a 3′-UTR fragment of CBX5 with the predicted binding site or a mutant variant was prepared based on previous research (11). Luciferase activities were measured 48 h after transfection using the dual-luciferase reporter assay kit (Vazyme, Jiangsu, China). Firefly luciferase activity was normalized to *Renilla* luciferase activity for each sample. sRNA45033 mimic and inhibitor (reverse complement to mimic) were synthesized by Ribobio (Guangzhou, China).

**CUT&Tag analysis, library construction, and DNA sequencing.** The CUT&Tag library was prepared using the Hyperactive Universal CUT&Tag assay kit for Illumina (Vazyme, Jiangsu, China) according to the manufacturer's instructions. A total of 100,000 cells were harvested, washed, and mixed with activated concanavalin A-coated magnetic beads at room temperature for 10 min. The mixture was resuspended in 50 $\mu$L antibody buffer consisting of primary antibody (1:50 dilution) and incubated overnight at 4℃. After being washed, 100 $\mu$L of pA/G-Tnp adapter complex ($\sim$0.04 $\mu$M) was added to the cells and incubated at room temperature for about 1 h. After washing, the cells were resuspended in Tagmentation buffer (50 $\mu$L) and incubated at 37℃ for 1 h. Then, proteinase K treatment and DNA extract beads (Vazyme, Jiangsu, China) were used at 55℃ for 10 min, and DNA was eluted. For PCR, the following thermocycler program was used: 72℃ for 3 min, followed by 95℃ for 3 min, 10 cycles of 98℃ for 10 s and 60℃ for 5 s, final extension at 72℃ for 1 min, and hold at 4℃. Pooled libraries were purified with VAHTS DNA clean beads (Vazyme, Jiangsu, China). Sequencing was performed with an Illumina NovaSeq 6000 platform (provided by Annoroad Company, Beijing, China), with a sequencing depth of 6 Gb for each sample. The data were visualized using Integrative Genomics Viewer and Vazyme cloud services.

**Statistical analysis.** The results are expressed as means $\pm$ standard deviation (SD). Experiments were repeated independently at least three times. Statistical significance was assessed with Student's *t* test or analysis of variance (ANOVA) using SPSS software and GraphPad Prism 5. A *P* value of <0.05 was considered statistically significant.

**Data availability.** Data sets have been uploaded to the Gene Expression Omnibus under GEO accession no. GSE218606.

## SUPPLEMENTAL MATERIAL

Supplemental material is available online only.
**SUPPLEMENTAL FILE 1**, PDF file, 7.3 MB.

## ACKNOWLEDGMENTS

We thank the Jiangsu Province Key Laboratory of Oral Diseases and the laboratory staff. We also thank these organizations for their financing.

This work was supported by the National Natural Science Foundation of China (grant no. 82170962, 81771074), the Key Projects of Social and Development of Jiangsu Department of Science and Technology (grant no. BE2020707), the Priority Academic Program Development of Jiangsu Higher Education Institutions (grant no. PAPD-2018-87), the Project of Cadre Health of Jiangsu Commission of Health (grant/award no. BJ19033), and the Project of Jiangsu Provincial Health and Family Planning Commission (grant/award no. H2018044).

We declare no conflict of interest.

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
