## [Reviewer comments · Microbiology Spectrum]

Microbiology Spectrum

***Porphyromonas gingivalis* outer membrane vesicles promote apoptosis via msRNA-regulated DNA methylation in periodontitis**

ruyi Fan, Yi Zhou, Xu Chen, Xianmei Zhong, Fanzhen He, Wenzao Peng, Lu Li, xiaoqian wang, and yan xu

Corresponding Author(s): yan xu, Department of Periodontics, Affiliated Hospital of Stomatology, Nanjing Medical University,

Review Timeline:

Submission Date:	August 19, 2022
Editorial Decision:	September 30, 2022
Revision Received:	November 28, 2022
Accepted:	December 7, 2022

Editor: John Atack

Reviewer(s): Disclosure of reviewer identity is with reference to reviewer comments included in decision letter(s). The following individuals involved in review of your submission have agreed to reveal their identity: Ling Ning Kaylie Lam (Reviewer #4)

Transaction Report:

DOI: <https://doi.org/10.1128/spectrum.03288-22>

September 30, 2022

Dr. yan xu
Department of Periodontics, Affiliated Hospital of Stomatology, Nanjing Medical University,
136 Hanzhong Road, Nanjing
nanjing
China

Re: Spectrum03288-22 (*Porphyromonas gingivalis* outer membrane vesicles promote apoptosis via msRNA-regulated DNA methylation in periodontitis)

Dear Dr. yan xu:

Link Not Available

Sincerely,

John Atack

Journals Department
Editor comments:
Ensure all NGS data is deposited in a suitable public repository.

Reviewer comments:

Reviewer #1 (Public repository details (Required)):

There are number of large data sets (which are impossible to interpret as the molecules can not be read) should be deposited into a public domain.

Reviewer #1 (Comments for the Author):

One of the major shortcomings of this manuscript is that there is no strong physiologically relevant data for the proposed role of

the vesicles in periodontitis as the clinically relevant data largely lacks to show that these vesicles are indeed secreted or shed in such high concentrations (utilized in this study) in human. The used methods and in vivo model also present key limitations and confounding factors to robustly interpret the results and their significance. A number of figures are not in the acceptable resolution to see them which make it impossible to understand them. The study is not very original as a similar study has been published already in 2017 (see the specific comments below). The manuscript is excessively long and look very descriptive. The noted issues stand beyond major revision.

Major Specific Comments:

1. Line 329: The ligature-based periodontitis causes a large induction in the localized levels of pro-inflammatory cytokines, please provide justification or references for your findings which state that the pro-inflammatory and pro-apoptotic responses are due to treatment with only the *P. gingivalis* OMVs.
2. The process for silver staining was never defined in the materials and methods. Additionally, it would have been more advantageous to utilize a *P. gingivalis* specific antibody, in order to confirm the presence of *P. gingivalis* related effector molecules in the OMVs.
3. Figure 1: Please either show more of the purified OMVs via TEM, or attempt to utilize another format to do so, such as SEM, to provide greater context to the vesicular morphology.
4. Figure 2: Use of *P. gingivalis* live bacterial cells is missing in this experimental design and in other experiments where functional studies of OMVs were performed. Based on literature, both *Pseudomonas aeruginosa* and *P. gingivalis* contains two different types of LPS. Therefore, OMVs of *P. aeruginosa* could be a better control to justify that resorption of alveolar bone observed in this study was specific to molecules present in OMVs of *P. gingivalis* not to LPS or not OMVs of any other bacteria which are not associated with periodontitis like *P. aeruginosa*.
5. It is prudent to establish the infection model of hPDLs with live *P. gingivalis* bacteria first and that live bacteria must be included in all functional assays. Moreover, two important pro-apoptotic proteins NOXA and PUMA which are direct target of p53 in apoptosis pathway has not been included in this study. Interestingly, Ref. 17 (Choi et al 2017) did similar and more comprehensive study with OMVs of periodontal pathogens including *P. gingivalis* and they found same sRNA45033 (*P.G_45033*) was one of the highly expressed sRNA in *P. gingivalis*' OMVs. In their study, they directly used this sRNA to investigate the expression of 16 cytokines including IL-1 β and IL-6. But they did not find any change in expression of IL-1 β and IL-6. However, this study showed higher expression of these two cytokines. The authors should explain this contradictory finding in their assays.
6. Figure 4: It is impossible to read any labels in Fig. 4A-F. So, they are not interpretable unless the quality of the image has been improved significantly. Fig. 4H appears to be the qRT-PCR result of 6 possible target genes of the host. So, which one is then further validated qRT-PCR result (please see lines 153-160 and correct it). Based on the Fig. 4G, expression of s45033 RNA ranked 5th among the highly expressed sRNAs and its expression was not so significant compared to other highly expressed 4 sRNAs. Then question arises, why this particular s45033 RNA was selected?
7. Figure 5: Specify what inhibitor was used for sRNA45033? It is necessary to show by fluorescence staining that sRNA45033 was successfully invaded/taken up by the hPDLs. Please see Ref. 17 (Choi et al 2017) for helpful tips.
8. Figure 6: Why was only PI stained utilized in the immunofluorescence work, when both p53 and Bcl-2 could be visualized and would be more effective evidence?
9. Figure 7: Densitometric analysis of western blot results of at least three biological replicates should be included. Lines 198-208: please clarify the relationship between p53 methylation and H3K9me3 (Histone 3 Lys9 trimethylation).
10. More background information with mechanistic studies on noncoding small RNAs (which is incorrectly mentioned in the manuscript as msRNAs) found in *P. gingivalis* OMVs and/or other bacteria should be included.
11. It appears that most of the citations, if not all; are not matching with the context in the manuscript.

Moderate to Minor Comments:

1. Line 16: Abstract is misspelled.
2. Line 18: replace substances with bioactive molecules.
3. Line 21: "its participation" should be "their participation"
4. Line 21 and throughout the manuscript: in vivo and in vitro should be italicized
5. Line 23: A sentence should not start with "And".
6. Line 24: Define what hPDLs are when mentioning the cell line for the first time and it is applicable for any acronyms when they are used for the first time. Is it human periodontal ligament cells?
7. Line 26, 69, and throughout the manuscript: what does mean by ms in msRNA? Did you mean it as medium-sized RNAs? Then the size mentioned in line 69 as 15-25 does not qualify to be medium-sized (which is 50-300 nucleotides: <https://pubmed.ncbi.nlm.nih.gov/33404286/>). Therefore, the title of the manuscript needs to be corrected accordingly.
8. Line 27: please specify screening of what?
9. Line 31: This sentence does not make sense as it is stated. Do you mean "... confirmed that CBX5 regulates apoptosis through the methylation of p53 DNA"?
10. Lines 35-39: "This study.....periodontitis" could be reframed in a 1) and 2) listing of the findings.
11. Line 41: O in Outer should be capitalized.
12. Line 46: This sentence should have a comma between "...chronic inflammation, which..."
13. Line 47 and throughout the manuscript: g in gram-negative should be capitalized.
14. Line 48: This sentence should have a comma between "...periodontal homeostasis, which..."
15. Line 50: Use the term "dysbiosis" instead of "infection", as the bacterial species are already present.

16. Line 53: Start the sentence with "The..."
17. Lines 54 and 211-212: none of the places correct reference (<https://www.ncbi.nlm.nih.gov/pmc/articles/PMC3498498/>) for the keystone pathogen has been cited.
18. Line 59: There should be a "the" between "from" and "bacterial"
19. Line 61: Again, wrong reference (Ref. 11) is cited for the size of the OMV of *P. gingivalis*. Ref 11 is a review paper but did not mention about the exact size of any Gram-negative bacterial vesicles. Liu et al (2021) *Mol Oral Microbiol* 36:182-191 paper mentioned the size of *P. gingivalis* OMVs as 50-400 nm. This paper is cited in this manuscript (Ref. 44) but in different context.
20. Line 65: Please clarify what do you mean by "Genetic materials (DNAs and RNAs) are also detected in the periodontal pathogens"? as all bacteria have DNA and RNA.
21. Line 72: The term "done" is too informal. Please use "performed".
22. Line 72: A comma needs to be inserted between "roles" and "which".
23. Line 76: replace binding with bind.
24. Line 77: Please add a "the" before the "trimethylated"; also, please specify the lysine residue.
25. Line 79: Please specify what are effects of CBX5 on inflammatory response genes.
26. Line 81: Again, wrong reference (Ref. 22). This paper (Ref. 22) did not discuss about p53 at all.
27. Line 83: Please add a comma between "hPLDCs" and "subsequently".
28. Lines 83 to 85: This sentence is very long and confusing to read, please separate it into two sentences.
29. Line 88: Please change the phrasing to "... between host, pathogen, and OMVs plays..."
30. Line 93: Please remove 'a'.
31. Line 97: Add protein in between several and bands. Please indicate the size of the protein marker used in Fig. 1D.
32. Line 101: Delete "further". Also define SD as it is used for the first time here. As mentioned above it is applicable for all acronyms.
33. Line 103: Does "the OMVs" mean the OMVs-treated group?
34. Line 104: Again, define what CP means.
35. Line 108: "...surface..." should be "...surfaces..."
36. Line 109: "...than..." should be "...when compared to..."
37. Line 126: There should be a comma between "...which..." and "...suggested..."
38. Line 132: This line should be "... in cells treated with..."
39. Line 133: What is meant by "On the other hand..."?
40. Line 136 and throughout the manuscript: it would be very unfair to generalize p53 as pro-apoptotic protein since it is undoubtedly known as a hallmark regulator of tumor-suppressor protein. So, delete pro-apoptotic protein part.
41. Line 137: There should be a "when" before "compared".
42. Line 140: Please provide the specifications for the purchased kit, such as the producing company.
43. Line 143: Define NLRP3 and briefly explains its roles in apoptosis/inflammation.
44. Line 148: Delete repeated words "gene regulation".
45. Line 149: Ref. 16 is also wrong citation here.
46. Line 156: This line is poorly phrased, please revise it to make it flow better.
47. Line 158: the phrase should be "lowly expressed"
48. Line 164: Please provide references for how the two constructed plasmid were created.
49. Line 165: "while" should instead be either "and" or "but."
50. Line 167: This sentence is a run-on. Please split it into two sentences.
51. Line 173: This line is poorly phrased, please revise it to make it flow better.
52. Line 173: It should be "...the apoptosis marker..." as p53 is only one marker.
53. Line 199: The phrase should be "... the mechanism for how..."
54. Line 202: This sentence's introduction is oddly phrased. Try instead, "Following the selective of hPDLCs with *P. gingivalis* OMVs..."
55. Line 206: Please define more clearly what is being treated with the *P. gingivalis* OMVs.
56. Line 212: "...which is evidenced by numerous studies" does not need to be said. Additionally, do not start the next sentence with a conjunction.
57. Line 215: Instead of "have", use the work "form".
58. Line 220: Add the word "dysfunction" after the word "joint".
59. Line 231: This sentence is a run-on. Please separate it.
60. Line 233: What is meant by the word "technic"?
61. Line 241: This should be "... aberrant ROS levels and showed signs..."
62. Line 242: Do not start a sentence with the word "And...". Additionally, the phrasing should be "NLRP3 and the pro-inflammatory cytokines..."
63. Line 243: The phrasing should be "...was significantly higher when hPDLCs were treated with..."
64. Line 244: Add a "and" before you discuss the expression changes of the apoptosis-related proteins.
65. Line 248: This sentence is phrased poorly, please revise.
66. Line 254: This sentence is phrased poorly, please revise.
67. Line 265: Please add additional citations for this sentence, as this is well characterized and has been examined in detail.
68. Line 270: The phrase should be "... a specific antibody binds to a chromatin protein..."
69. Line 277: The word should be "underwent", rather than "undertook".
70. Line 282: Do not start a sentence with "And".

71. Line 287: The end of this sentence doesn't really work; try instead "... and identify their effectiveness as possible treatment targets."
72. Line 288: This sentence is a run-on. Please separate it.
73. Line 292: Add a comma after the word "host cells"
74. Line 298: Please provide references that prove the efficacy of your culturing techniques for producing viable *P. gingivalis* growth.
75. Line 300: Is there any particular reason why 10% sheep blood was used as almost all published papers on *P. gingivalis* used 5%. One more thing, it is defibrinated, not defibrated.
76. Line 302-303: please be consistent with which one was true, jar or chamber?
77. Line 307: how is it possible to have OD so low about one even after more than 48 h of growth?
78. Line 311: what does mean by (-) in PBS (-).
79. Line 329: add chronic before periodontitis.
80. Line 336: How was the nM concentration determined as it needs Molecular Weight (MW) for molar concentration calculation, and it would be interesting to know how was the MW of OMVs determined?
81. Line 474: Please attempt to maintain a singular tense. Additionally, the term "should" should not be utilize as it is expected that these methodologies were performed.
82. Figure 1: The bands of Figure 1D, especially when comparing the bands of the *P. gingivalis* OMVs to free *P. gingivalis*, look very blurred and dissimilar. A repeated picture is necessary.
83. Figure 3: Please increase the intensity of the red in Figure 3A and 3C, as it is very difficult to see the Dil-labelled OMVs and the PI stain, respectively.
84. Figure 5: Please increase the intensity of the red in Figure 5F, as it is very difficult to see the PI stain.
85. Figure 6: Please increase the intensity of the red in Figure 6F, as it is very difficult to see the PI stain.

Reviewer #3 (Public repository details (Required)):

The authors performed transcriptome sequencing analysis of *Porphyromonas gingivalis* outer membrane vesicles.

Reviewer #3 (Comments for the Author):

I reviewed the article entitled "Porphyromonas gingivalis outer membrane vesicles promote apoptosis via msRNA-regulated DNA methylation in periodontitis" by Fan et al. The manuscript addresses an important gap in knowledge about the link between *Porphyromonas gingivalis* and periodontal disease. The authors use in vitro and in vivo experiments. My comments to improve the manuscript can be found below:

Abstract

Line 24: "hPLDCs" should be "hPDLCs". Please correct it and revise this throughout the manuscript.

Lines 26-29: The sentence in these lines is confusing (it seems like the sRNA 45033 regulates apoptosis in hPDLCs using a dual-luciferase reporter assay, overexpression and knockdown methods). Please re-write it.

Introduction:

Line 76: "binding" should be "binds".

Lines 95-97: The authors mention in line 61 that *P. gingivalis* OMVs have a size of around 100nm. In the results of this manuscript (Figure 1C), the authors report isolation of OMVs with an average of 202.8 +71.87 nm (ranging from 100 to about 500nm). How do the authors explain that the size of the OMVs isolated by them are greater than the 100nm size previously reported? Is there a difference in OMVs size according to different *P. gingivalis* strains?

Line 309: The authors describe that the supernatants from *P. gingivalis* is filtered using a 0.22 μ m syringe filter before the samples are centrifuged at high speed to concentrate the OMVs. In figure 1C, the authors show that their OMVs vary in size (ranging from 100 to about 500nm) with an average size of 202.8+71.87 nm. How can the authors obtain OMVs that are larger than the pore of the filter (200nm)? Is it possible that the OMVs fuse after the concentration step? This point should be discussed and added to the manuscript.

Methods: The authors should state, at least when it is first cited, what are hPDLCs.

The authors mention that the primary hPDLCs were isolated from human samples. How did the authors confirm that the cells in the manuscript were hPDLCs? Are there cell markers to detected by the authors that confirmed that the cells used in this study were hPDLCs?

Results:

Fig 1A: The authors should add in the image that the chronic periodontitis control group received PBS injections (instead of OMVs). Is that correct to assume?

What was the volume used to inject PBS or OMVs to rats' mouths?

Line 113: The authors should explain what "Dil-labeled *P. gingivalis* OMVs" is, in the results section. Or at least refer the readers to the Methods section.

Figure 3: How much of the OMVs were used in the experiments? The authors show a time-dependent induction of cell death by OMVs, but they do not show if OMVs induce cell death in low or high concentrations/amounts. I recommend that the authors add results on the different concentrations/ amounts of OMVs that can induce cell death of hPDLCs.

Figure 3A: The authors show, by immunofluorescence microscopy, that there is co-localization of OMVs (in red) with hPLDCs (in green). To this reviewer, this experiment shows co-localization instead of internalization of OMVs. Can the authors show internalization of OMVs by hPLDCs using a different technique? Confocal microscopy? Z-stack images/videos? Flow cytometry? Hence, Figure 3A has an additional text under the word "merge". It is unclear for me what the text means.

Figure 3G: The authors should include the densitometric analyses to western blot images in Figure 3G.

Figure 4A and 4D: The data in these images are too small and impossible to read. The authors should provide better quality images, or provide the information as supplementary data, if they don't fit in the main text of the manuscript.

Figure 5: The experiment conducted in Fig5B needs to be better explained in the results section.

Are the data on the transcriptome sequencing analysis of *Porphyromonas gingivalis* OMVs available in a public repository?

Reviewer #4 (Comments for the Author):

The manuscript entitled "Porphyromonas gingivalis outer membrane vesicles promote apoptosis via msRNA-regulated DNA methylation in periodontitis" by Xu et al. reports that outer membrane vesicles (OMVs) produced by *P. gingivalis* stimulates alveolar bone resorption and apoptosis of hPLDCs. Furthermore, they uncovered that apoptosis of hPLDCs is driven by OMV-derived sRNA45033 targeting eukaryotic Chromobox 5 (CBX5) that regulates p53 and cell apoptosis. Overall, this study advances the field of *P. gingivalis* pathogenesis and its role in periodontitis; however, it is of my opinion that several important questions remain to be clarified and this manuscript can be further improved.

Major comments:

- Since the OMVs isolated had a broad distribution in terms of size, do the authors think that size differences in OMVs can impact the results in the experiments performed? Additionally, can the author comment on whether OMVs size composition/population differ among biological replicates?
- The authors reasoned that OMV-derived sRNA45033 is targeting CBX5. Did the authors consider using OMVs isolated from a mutant strain lacking sRNA45033 as a negative control? Although the use of CBX5 wild-type (pCBX5-WT) and mutant plasmid (pCBX5-Mut), and its inhibitor indeed validated the role of sRNA45033, I wondered to what degree sRNA45033 causes cell apoptosis. Is it possible that there are other redundant sRNA?
- It is interesting that PG-derived sRNA45033 target hPLDCs for apoptosis. Did the authors perform a phylogenetic analysis to determine whether this is a conserved mechanism for PG, or at genus level or in gram negative pathogens? I would recommend including this analysis into the main figures.
- For Fig 1, did the authors perform normalization and/or separation of OMV based on size? It would be beneficial to show the distribution/probability of the size of OMV harvested from different biological samples to account for biological variability. Assuming the author is making a statement on protein content between OMV and *P. gingivalis*, the OMV and PG cell lysate has to be normalized by protein content which is not seen in Fig 1D. My recommendation is to remove Fig 1D or move to supplementary.
- In Fig 2, the authors reasoned that the hematoxylin (staining nuclei cell purple) and eosin (staining extracellular matrix and cytoplasm pink) stained images support the notion that OMV altered bone resorption, which I do not agree. H&E staining is relatively crude and is not specific to a cell lineage. Further, the H&E image is not quantitative data, but qualitative hence it is not possible to demonstrate significance. Similarly, TRAP stains tartrate-resistant acid phosphatase which is present in osteoclast; however, these enzymes are also present in immune cells too. I would suggest the authors rephrase their interpretations of the result. Alternatively, performing flow cytometry with specific cell markers should suffice to validate their reasoning.

Minor comments:

- Throughout the manuscript, please italicize *in vivo* and *in vitro*
- The first use of the abbreviation hPLDCs appeared in the abstract, yet there is no mention what is hPLDCs. Please clarify the rationale for using hPLDCs.
- In line 68, please reference the study.
- In line 71, please provide some information on the downstream genes regulated by *P. gingivalis* msRNA.
- In line 76, change "binding" to "bind"
- In line 77, change "trimethylated" to "tri-methylated"
- In line 79, please clarify on the specific function of CBX5.
- In line 148, "gene regulation" was mentioned twice.
- Fig 4A-F images are too small to be seen. I would recommend moving 4B-D to supplementary, and combine 4E-F together into a bigger figure. This would make room for 4A to be large too. Please provide more information on these figures in the figure legend.
- Please clarify what analysis was performed to determine CBX5 quantification shown in Fig 4K, and what staining are done in Fig 4J.
- Please clarify the sentence in line 164 "The s45033 mimics significantly decreased the luciferase activity of the wild-type (WT) while did not affect the mutant". I would suggest renaming it to s45033 recombinant WT (r-WT) as the sentence sounds confusing.

- In line 168, the authors mentioned "inhibition of s45033 dramatically reversed 168 cell viability". Can the authors explain the mechanism of inhibition that is happening here?

Staff Comments:

Preparing Revision Guidelines

Please return the manuscript within 60 days; if you cannot complete the modification within this time period, please contact me. If you do not wish to modify the manuscript and prefer to submit it to another journal, please notify me of your decision immediately so that the manuscript may be formally withdrawn from consideration by Microbiology Spectrum.

The article entitled “*Porphyromonas gingivalis* outer membrane vesicles promote apoptosis via msRNA-regulated DNA methylation in periodontitis” demonstrates the *in vitro* and *in vivo* effects of *Porphyromonas gingivalis* OMVs in inducing apoptosis in periodontitis. The Authors constructed the study and represented accordingly. However, a few points are to be taken into consideration to enrich with more factual information in the line of the topic. My comments have been listed below:

Major comments:

1. Line no. 67-70, page no. 4, “Indeed, a recent study has revealed that OMVs could deliver a novel class of small RNAs (msRNAs, 15-25 nucleotides) into the host,in the meantime” Please cite the reference. Mention the target of OMVs delivery.
2. Line no. 305-318, page no. 15-16, “Isolation and identification of *P. gingivalis* OMVs”. Authors have taken the death phase culture of *P. gingivalis* for isolating OMVs. Death phase culture contains high amount of lysed bacterial products/remnants contaminations, which can be found in the culture supernatant even after filtration (0.22- μm) and ultracentrifugation (100000 \times g). This may effect/ influence the test result.
3. Authors are requested to check the OMVs’ sterility by culturing on plates.
4. Please mention the count of OMVs (particles/mL) mixed with cell culture for pretreatment.
5. Fig. 1D, please mention the molecular weight of protein ladder. Also to be mentioned the amount of proteins loaded in each well of SDS-PAGE.
6. Fig. 2D, what is indicating by red arrows? Please mention in the legend.

7. Fig. 3I, TNF- α is not showing significant change with OMVs. How can authors explain it? Please write in the discussion.

Minor Comments:

1. Fig. 1B, scale is not visible properly. Please mention scale in white.
2. Fig. 2A, E, F, please increase font size and make the fonts more clear to read.
3. Reference No-34, 45 please rewrite as per the Journal's standard format.

Response to Reviewers

Responses to Reviewer #1

Thank you for your detailed comments and suggestions. We found them quite useful as we approached our revision. We tried our best to improve the manuscript and made corresponding revisions.

Reviewer #1 (Public repository details (Required)):

There are number of large data sets (which are impossible to interpret as the molecules can not be read) should be deposited into a public domain.

Response: Thank you for pointing out this oversight. NGS data sets have been deposited into GEO (accession number GSE218606). (Line 179-180)

Reviewer #1 (Comments for the Author):

Point 1: Line 329: The ligature-based periodontitis causes a large induction in the localized levels of pro-inflammatory cytokines, please provide justification or references for your findings which state that the pro-inflammatory and pro-apoptotic responses are due to treatment with only the *P. gingivalis* OMVs.

Response: Thank you for your thorough review and salient observations. In the animal study, we demonstrated that the ligature-based periodontitis and *P. gingivalis* OMVs treatment both collaboratively and separately caused significant bone resorption. In the latter studies, we confirmed that *P. gingivalis* OMVs treatment alone could induce pro-apoptotic responses (Flow cytometry, caspase-3 activity, and expression of Bcl-2 and

Bax) and pro-inflammatory response (ELISA). Here references stated that *P. gingivalis* OMVs induce inflammatory responses (1, 2). Unfortunately, there aren't many references related to *P. gingivalis* OMVs and apoptosis, however there is a recent study showing that the OMVs produced by Gram-negative bacteria activate apoptosis and inflammation. And we think a lack of study in this aspect is one of the hallmarks of our findings.

Reference:

1. Uemura Y, Hiroshima Y, Tada A, Murakami K, Yoshida K, Inagaki Y, Kuwahara T, Murakami A, Fujii H, Yumoto H. 2022. Porphyromonas gingivalis Outer Membrane Vesicles Stimulate Gingival Epithelial Cells to Induce Pro-Inflammatory Cytokines via the MAPK and STING Pathways. Biomedicines 10:2643.
2. Cecil JD, O'Brien-Simpson NM, Lenzo JC, Holden JA, Singleton W, Perez-Gonzalez A, Mansell A, Reynolds EC. 2017. Outer Membrane Vesicles Prime and Activate Macrophage Inflammasomes and Cytokine Secretion In Vitro and In Vivo. Front Immunol 8:1017.

Point 2: The process for silver staining was never defined in the materials and methods. Additionally, it would have been more advantageous to utilize a *P. gingivalis* specific antibody, in order to confirm the presence of *P. gingivalis* related effector molecules in the OMVs.

Response: Thank you for this instructive suggestion. We've defined the silver staining in the material and methods. (Line 368-372)

The authors recognize the reviewer's concern. In this study, our primary target was sRNA hence we didn't provide *P. gingivalis*-related effector (lipopolysaccharide, gingipains, pili) evidence. Still, the suggestion is valuable and we'll utilize *P. gingivalis*-specific antibodies such as gingipain antibodies in future research.

Point 3: Figure 1: Please either show more of the purified OMVs via TEM, or attempt to utilize another format to do so, such as SEM, to provide greater context to the vesicular morphology.

Response: We thank you for your constructive suggestions. We've extracted and purified the OMVs aging and subjected them to TEM and SEM. We've incorporated more images of *P. gingivalis* OMVs taken by TEM in Figure.1. Due to the time limitation, we have not acquired any good-quality SEM images yet. We'll keep trying to obtain high-quality SEM images along with our future study.

Point 4: Figure 2: Use of *P. gingivalis* live bacterial cells is missing in this experimental design and in other experiments where functional studies of OMVs were performed. Based on literature, both *Pseudomonas aeruginosa* and *P. gingivalis* contains two different types of LPS. Therefore, OMVs of *P. aeruginosa* could be a better control to justify that resorption of alveolar bone observed in this study was specific to molecules present in OMVs of *P. gingivalis* not to LPS or not OMVs of any other bacteria which are not associated with periodontitis like *P. aeruginosa*.

Response: Thank you for your detailed comments and suggestions. We designed the experiments based on previous literature that used PBS or *P. gingivalis* LPS as control (1–3). To avoid the possibility of live bacterial cells producing OMVs in the experimental process, we chose the *P. gingivalis* LPS as a control. But we agree with the reviewer that *P. aeruginosa* could serve as a better control. We'll acquire this strain and its LPS for future study.

Reference:

1. Uemura Y, Hiroshima Y, Tada A, Murakami K, Yoshida K, Inagaki Y, Kuwahara T, Murakami A, Fujii H, Yumoto H. 2022. Porphyromonas gingivalis Outer Membrane Vesicles Stimulate Gingival Epithelial Cells to Induce Pro-Inflammatory Cytokines via the MAPK and STING Pathways. Biomedicines 10:2643.
2. Silva IL, Cascales E. 2021. Molecular Strategies Underlying Porphyromonas gingivalis Virulence. J Mol Biol 433:166836.
3. Deo P, Chow SH, Han M-L, Speir M, Huang C, Schittenhelm RB, Dhital S, Emery J, Li J, Kile BT, Vince JE, Lawlor KE, Naderer T. 2020. Mitochondrial dysfunction caused by outer membrane vesicles from Gram-negative bacteria activates intrinsic apoptosis and inflammation. Nat Microbiol <https://doi.org/10.1038/s41564-020-0773-2>.

Point 5. It is prudent to establish the infection model of hPDLs with live *P. gingivalis* bacteria first and that live bacteria must be included in all functional assays. Moreover, two important pro-apoptotic proteins NOXA and PUMA which are direct target of p53 in apoptosis pathway has not been included in this study. Interestingly, Ref. 17 (Choi et al 2017) did similar and more comprehensive study with OMVs of periodontal pathogens including *P. gingivalis* and they found same sRNA45033 (P.G_45033) was one of the highly expressed sRNA in *P. gingivalis*'

OMVs. In their study, they directly used this sRNA to investigate the expression of 16 cytokines including IL-1 β and IL-6. But they did not find any change in expression of IL-1 β and IL-6. However, this study showed higher expression of these two cytokines. The authors should explain this contradictory finding in their assays

Response: We thank you for your constructive suggestions which have led to a stronger and clearer revised manuscript. Because the infection model of hPDLCs with live *P. gingivalis* bacteria has been documented (1, 2) and *P. gingivalis* LPS has been used in experimentally induced periodontitis (3–5), we included the results of *P. gingivalis* as control. But as mentioned before, we agree with the reviewer that *P. aeruginosa* LPS could be a better control in the experiments. We'll acquire *P. aeruginosa* LPS and further study this matter in detail.

The expression of pro-apoptotic proteins NOXA and PUMA has been measured and included in the study and the manuscript has been revised. (Line 162-164, 209-211, 231-232)

Fig. S4

Figure. S4 Western Blot and the densitometric analyses of PUMA and NOXA expression. $*p < 0.05$.

The cytokines expression was utilized as a functional assay of our purified OMVs. The difference between Ref. 17 (Choi et al 2017) is that they directly transfected this sRNA into Jurkat T cells as we subjected OMVs to the hPDLCs. The contradictory finds may be caused by different cell lines. Another hypothesis is s45033 may not necessarily link to these 2 cytokines. We thank the author for pointing out this oversight and we'll further look into this matter in detail.

Reference:

1. Liu J, Duan J, Wang Y, Ouyang X. 2014. Intracellular adhesion molecule-1 is

regulated by porphyromonas gingivalis through nucleotide binding oligomerization domain-containing proteins 1 and 2 molecules in periodontal fibroblasts. *J Periodontol* 85:358–368.

2. Liu J, Wang Y, Ouyang X. 2014. Beyond toll-like receptors: Porphyromonas gingivalis induces IL-6, IL-8, and VCAM-1 expression through NOD-mediated NF- κ B and ERK signaling pathways in periodontal fibroblasts. *Inflammation* 37:522–533.
3. Yang F, Huang D, Xu L, Xu W, Yi X, Zhou X, Ye L, Zhang L. 2021. Wnt antagonist secreted frizzled-related protein I (sFRP1) may be involved in the osteogenic differentiation of periodontal ligament cells in chronic apical periodontitis. *Int Endod J* 54:768–779.
4. Blufstein A, Behm C, Nguyen PQ, Rausch-Fan X, Andrukhov O. 2018. Human periodontal ligament cells exhibit no endotoxin tolerance upon stimulation with Porphyromonas gingivalis lipopolysaccharide. *J Periodontal Res* 53:589–597.
5. Li X, Yu C, Hu Y, Xia X, Liao Y, Zhang J, Chen H, Lu W, Zhou W, Song Z. 2018. New Application of Psoralen and Angelicin on Periodontitis With Anti-bacterial, Anti-inflammatory, and Osteogenesis Effects. *Front Cell Infect Microbiol* 8:178.

Point 6. Figure 4: It is impossible to read any labels in Fig. 4A-F. So, they are not interpretable unless the quality of the image has been improved significantly. Fig,

4H appears to be the qRT-PCR result of 6 possible target genes of the host. So, which one is then further validated qRT-PCR result (please see lines 153-160 and correct it). Based on the Fig. 4G, expression of s45033 RNA ranked 5th among the highly expressed sRNAs and its expression was not so significant compared to other highly expressed 4 sRNAs. Then question arises, why this particular s45033 RNA was selected?

Response: Thank you for your thorough review and salient observations. Figure 4 has been rearranged and revised and line 153-160 (now line 185-189) has also been revised. After screening out the highly expressed sRNAs, we used bioinformatic analysis to predict the possible targets of these sRNAs, we found only the expression of CBX5 (the downstream of s45033) has changed significantly. This prompted us to decide to study s45033 first, and we're planning on exploring the bio-function of each highly expressed sRNA in the future.

Point 7. Figure 5: Specify what inhibitor was used for sRNA45033? It is necessary to show by fluorescence staining that sRNA45033 was successfully invaded/taken up by the hPDLCs. Please see Ref. 17 (Choi et al 2017) for helpful tips.

Response: Thank you very much for your valuable suggestions. The manuscript has been revised to specify the sRNA45033 inhibitor. (Line 544-545)

We've produced confocal microscopy images regarding the sRNA45033 invading the hPDLCs. However, only synthetic sRNA45033 was subjected to hPDLCs to confirm this process. We're planning to use *in situ* hybridization technique to visualize the

process of specific sRNA as sRNA45033 enclosed by *P. gingivalis* while taken up by hPDLs.

sRNA45033 taken up by the hPDLs. DAPI(blue) and RNA-specific dye, SYTO RNASelect (green)

Point 8. Figure 6: Why was only PI stained utilized in the immunofluorescence work, when both p53 and Bcl-2 could be visualized and would be more effective evidence?

Response: Thank you for this instructive suggestion. PI staining was utilized to determine the cell viability after various stimulation in the manuscript. And the expressions of p53, NOXA, PUMA, and Bcl-2 were measured by Western Blot, which served as evidence in the apoptosis aspect.

Point 9. Figure 7: Densitometric analysis of western blot results of at least three biological replicates should be included. Lines 198-208: please clarify the relationship between p53 methylation and H3K9me3 (Histone 3 Lys9 trimethylation).

Response: Thank you for your thorough review and instructive suggestion. The

densitometric analysis of western blot results has been included in the figures. The relationship has been clarified in the manuscript. (Line 235-237)

Point 10. More background information with mechanistic studies on noncoding small RNAs (which is incorrectly mentioned in the manuscript as msRNAs) found in *P. gingivalis* OMVs and/or other bacteria should be included.

Response: Thank you for this direction. The manuscript has been revised according to your suggestion and the background information has been added in the Introduction section. (Line 71-89)

Point 11. It appears that most of the citations, if not all; are not matching with the context in the manuscript.

Response: The authors appreciate the reviewer pointing out this oversight. The manuscript has been revised carefully. We apologize for whatever inconvenience this may cause you.

Moderate to Minor Comments:

Thank you for your thorough review and salient observations. Considering these suggestions, we tried our best to improve the manuscript and made many changes in the revised manuscript. We hope that the revisions made will meet with approval.

1. Line 16: Abstract is misspelled.

Response: Thank you for pointing out this oversight. The text has been revised. (Line 17)

2. Line 18: replace substances with bioactive molecules.

Response: Thank you for this direction. The text has been revised. (Line 19)

3. Line 21: "its participation" should be "their participation"

Response: Thank you for pointing out this oversight. The text has been revised. (Line 22)

4. Line 21 and throughout the manuscript: in vivo and in vitro should be italicized

Response: Thank you for pointing out this oversight. The text has been revised. (Line 22)

5. Line 23: A sentence should not start with "And".

Response: Thank you for pointing out this oversight. The text has been revised. (Line 25)

6. Line 24: Define what hPLDCs are when mentioning the cell line for the first time and it is applicable for any acronyms when they are used for the first time. Is it human periodontal ligament cells?

Response: Thank you for pointing out this oversight. The text has been revised. (Line

25-26)

7. Line 26, 69, and throughout the manuscript: what does mean by ms in msRNA?

Did you mean it as medium-sized RNAs? Then the size mentioned in line 69 as 15-25 does not qualify to be medium-sized (which is 50-300 nucleotides: <https://pubmed.ncbi.nlm.nih.gov/33404286/>). Therefore, the title of the manuscript needs to be corrected accordingly.

Response: Thank you for your thorough review and salient observations. msRNA was referred as microRNA-size, small RNA by several groups (1–5). We've revised the manuscript to avoid confusion. (Line 27-28)

Reference:

1. Choi J-W, Kim S-C, Hong S-H, Lee H-J. 2017. Secretable Small RNAs via Outer Membrane Vesicles in Periodontal Pathogens. *J Dent Res* 96:458–466.
2. Ahmadi Badi S, Bruno SP, Moshiri A, Tarashi S, Siadat SD, Masotti A. 2020. Small RNAs in Outer Membrane Vesicles and Their Function in Host-Microbe Interactions. *Front Microbiol* 11:1209.
3. Nejman-Faleńczyk B, Bloch S, Licznarska K, Dydecka A, Felczykowska A, Topka G, Węgrzyn A, Węgrzyn G. 2015. A small, microRNA-size, ribonucleic acid regulating gene expression and development of Shiga toxin-converting bacteriophage Φ 24B. *Sci Rep* 5:10080.
4. Kang S-M, Choi J-W, Lee Y, Hong S-H, Lee H-J. 2013. Identification of

microRNA-size, small RNAs in Escherichia coli. Curr Microbiol 67:609–613.

5. Lee H-J, Hong S-H. 2012. Analysis of microRNA-size, small RNAs in Streptococcus mutans by deep sequencing. FEMS Microbiol Lett 326:131–136.

8. Line 27: please specify screening of what?

Response: Thank you for your thorough review and salient observations. The text has been revised. (Line 29)

9. Line 31: This sentence does not make sense as it is stated. Do you mean "... confirmed that CBX5 regulates apoptosis through the methylation of p53 DNA"?

Response: Thank you for your thorough review and salient observations. The text has been revised. (Line 34)

10. Lines 35-39: "This study.....periodontitis" could be reframed in a 1) and 2) listing of the findings.

Response: Thank you for this direction. The text has been revised. (Line 38-42)

11. Line 41: O in Outer should be capitalized.

Response: Thank you for pointing out this oversight. The text has been revised. (Line 45)

12. Line 46: This sentence should have a comma between "...chronic inflammation, which..."

Response: Thank you for pointing out this oversight. The text has been revised. (Line 51)

13. Line 47 and throughout the manuscript: g in gram-negative should be capitalized.

Response: Thank you for pointing out this oversight. The text has been revised. (Line 52)

14. Line 48: This sentence should have a comma between "...periodontal homeostasis, which..."

Response: Thank you for pointing out this oversight. The text has been revised. (Line 53)

15. Line 50: Use the term "dysbiosis" instead of "infection", as the bacterial species are already present.

Response: Thank you for pointing out this oversight. The text has been revised. (Line 55)

16. Line 53: Start the sentence with "The..."

Response: Thank you for pointing out this oversight. The text has been revised. (Line

58)

17. Lines 54 and 211-212: none of the places correct reference (<https://www.ncbi.nlm.nih.gov/pmc/articles/PMC3498498/>) for the keystone pathogen has been cited.

Response: Thank you for pointing out this oversight. The reference has been corrected.

(Line 59)

18. Line 59: There should be a "the" between "from" and "bacterial"

Response: Thank you for pointing out this oversight. The text has been revised. (Line

64)

19. Line 61: Again, wrong reference (Ref. 11) is cited for the size of the OMV of *P. gingivalis*. Ref 11 is a review paper but did not mention about the exact size of any Gram-negative bacterial vesicles. Liu et al (2021) Mol Oral Microbiol 36:182-191 paper mentioned the size of *P. gingivalis* OMVs as 50-400 nm. This paper is cited in this manuscript (Ref. 44) but in different context.

Response: Thank you for pointing out this oversight. The text and reference have been revised. (Line 66)

20. Line 65: Please clarify what do you mean by "Genetic materials (DNAs and RNAs) are also detected in the periodontal pathogens"? as all bacteria have DNA

and RNA.

Response: Thank you for pointing out this oversight. The text has been revised. (Line 71-73)

21. Line 72: The term "done" is too informal. Please use "performed".

Response: Thank you for pointing out this oversight. The text has been revised. (Line 90)

22. Line 72: A comma needs to be inserted between "roles" and "which".

Response: Thank you for pointing out this oversight. The text has been revised. (Line 90)

23. Line 76: replace binding with bind.

Response: Thank you for pointing out this oversight. The text has been revised. (Line 94)

24. Line 77: Please add a "the" before the "trimethylated"; also, please specify the lysine residue.

Response: Thank you for pointing out this oversight. The text has been revised. (Line 95-96)

25. Line 79: Please specify what are effects of CBX5 on inflammatory response

genes.

Response: Thank you for this direction. The effects of CBX5 on inflammatory response genes has been specified. (Line 98-99)

26. Line 81: Again, wrong reference (Ref. 22). This paper (Ref. 22) did not discuss about p53 at all.

Response: Thank you for pointing out this oversight. The reference has been corrected. (Line 101)

27. Line 83: Please add a comma between "hPLDCs" and "subsequently".

Response: Thank you for pointing out this oversight. The text has been revised. (Line 104)

28. Lines 83 to 85: This sentence is very long and confusing to read, please separate it into two sentences.

Response: Thank you for pointing out this oversight. The sentence has been revised. (Line 103-105)

29. Line 88: Please change the phrasing to "... between host, pathogen, and OMVs plays..."

Response: Thank you for pointing out this oversight. The text has been revised. (Line 108)

30. Line 93: Please remove 'a'.

Response: Thank you for pointing out this oversight. The text has been revised. (Line 113)

31. Line 97: Add protein in between several and bands. Please indicate the size of the protein marker used in Fig. 1D.

Response: Thank you for this direction. The text and the figure have been revised. (Line 117)

32. Line 101: Delete "further". Also define SD as it is used for the first time here.

As mentioned above it is applicable for all acronyms.

Response: Thank you for this direction. The text has been revised. (Line 122-123)

33. Line 103: Does "the OMVs" mean the OMVs-treated group?

Response: Thank you for this direction. The text has been revised to clarify the group.

(Line 125)

34. Line 104: Again, define what CP means.

Response: Thank you for this direction. The acronym has been defined. (Line 126-127)

35. Line 108: "...surface..." should be "...surfaces..."

Response: Thank you for pointing out this oversight. The text has been revised. (Line 134)

36. Line 109: "...than..." should be "...when compared to..."

Response: Thank you for pointing out this oversight. The text has been revised. (Line 135)

37. Line 126: There should be a comma between "...which..." and "...suggested..."

Response: Thank you for pointing out this oversight. The text has been revised. (Line 151)

38. Line 132: This line should be "... in cells treated with..."

Response: Thank you for pointing out this oversight. The text has been revised. (Line 157)

39. Line 133: What is meant by "On the other hand..."?"

Response: Thank you for pointing out this oversight. The text has been revised. (Line 158)

40. Line 136 and throughout the manuscript: it would be very unfair to generalize p53 as pro-apoptotic protein since it is undoubtedly known as a hallmark regulator of tumor-suppressor protein. So, delete pro-apoptotic protein part.

Response: Thank you for this direction. The text has been revised. (Line 161)

41. Line 137: There should be a "when" before "compared".

Response: Thank you for pointing out this oversight. The text has been revised. (Line 161)

42. Line 140: Please provide the specifications for the purchased kit, such as the producing company.

Response: Thank you for this direction. The specifications have been detailed in the materials and methods. (Line 522)

43. Line 143: Define NLRP3 and briefly explains its roles in apoptosis/inflammation.

Response: Thank you for this direction. The NLRP3 has been defined and the text has been revised. (Line 168-171)

44. Line 148: Delete repeated words "gene regulation".

Response: Thank you for pointing out this oversight. The text has been revised. (Line 178)

45. Line 149: Ref. 16 is also wrong citation here.

Response: Thank you for pointing out this oversight. The text and reference have been revised. (Line 180)

46. Line 156: This line is poorly phrased, please revise it to make it flow better.

Response: Thank you for pointing out this oversight. The text has been revised. (Line 186-189)

47. Line 158: the phrase should be "lowly expressed"

Response: Thank you for pointing out this oversight. The text has been revised. (Line 188-189)

48. Line 164: Please provide references for how the two constructed plasmid were created.

Response: Thank you for pointing out this oversight. The reference has been provided in the material and methods section. (Line 541)

49. Line 165: "while" should instead be either "and" or "but."

Response: Thank you for pointing out this oversight. The text has been revised. (Line 194-196)

50. Line 167: This sentence is a run-on. Please split it into two sentences.

Response: Thank you for pointing out this oversight. The sentence has been revised.
(Line 194-198)

51. Line 173: This line is poorly phrased, please revise it to make it flow better.

Response: Thank you for pointing out this oversight. The sentence has been revised.
(Line 205-209)

52. Line 173: It should be "...the apoptosis marker..." as p53 is only one marker.

Response: Thank you for pointing out this oversight. The sentence has been revised.
(Line 207)

53. Line 199: The phrase should be "... the mechanism for how..."

Response: Thank you for pointing out this oversight. The sentence has been revised.
(Line 235)

**54. Line 202: This sentence's introduction is oddly phrased. Try instead,
"Following the selective of hPDLCs with *P. gingivalis* OMVs..."**

Response: Thank you for pointing out this oversight. The sentence has been revised.

(Line 239-240)

55. Line 206: Please define more clearly what is being treated with the *P. gingivalis* OMVs.

Response: Thank you for this direction. The sentence has been revised. (Line 241-242)

56. Line 212: "...which is evidenced by numerous studies" does not need to be said. Additionally, do not start the next sentence with a conjunction.

Response: Thank you for pointing out this oversight. The text has been revised accordingly. (Line 250)

57. Line 215: Instead of "have", use the word "form".

Response: Thank you for pointing out this oversight. The text has been revised. (Line 253)

58. Line 220: Add the word "dysfunction" after the word "joint".

Response: Thank you for pointing out this oversight. The text has been revised. (Line 258)

59. Line 231: This sentence is a run-on. Please separate it.

Response: Thank you for pointing out this oversight. The text has been revised. (Line 266-268)

60. Line 233: What is meant by the word "technic"?

Response: Thank you for pointing out this oversight. The text has been revised. (Line 274)

61. Line 241: This should be "... aberrant ROS levels and showed signs..."

Response: Thank you for pointing out this oversight. The text has been revised. (Line 282)

62. Line 242: Do not start a sentence with the word "And...". Additionally, the phrasing should be "NLRP3 and the pro-inflammatory cytokines..."

Response: Thank you for pointing out this oversight. The text has been revised. (Line 282-283)

63. Line 243: The phrasing should be "...was significantly higher when hPDLCs were treated with..."

Response: Thank you for pointing out this oversight. The text has been revised. (Line 290)

64. Line 244: Add a "and" before you discuss the expression changes of the apoptosis-related proteins.

Response: Thank you for pointing out this oversight. The text has been revised. (Line

290)

65. Line 248: This sentence is phrased poorly, please revise.

Response: Thank you for this direction. The sentence has been revised. (Line 295-297)

66. Line 254: This sentence is phrased poorly, please revise.

Response: Thank you for this direction. The sentence has been revised. (Line 301-302)

67. Line 265: Please add additional citations for this sentence, as this is well characterized and has been examined in detail.

Response: Thank you for this direction. The text and reference have been revised. (Line 314)

68. Line 270: The phrase should be "... a specific antibody binds to a chromatin protein..."

Response: Thank you for pointing out this oversight. The text has been revised. (Line 318-319)

69. Line 277: The word should be "underwent", rather than "undertook".

Response: Thank you for pointing out this oversight. The text has been revised. (Line 325)

70. Line 282: Do not start a sentence with "And".

Response: Thank you for pointing out this oversight. The text has been revised. (Line 330)

71. Line 287: The end of this sentence doesn't really work; try instead "... and identify their effectiveness as possible treatment targets."

Response: Thank you for pointing out this oversight. The text has been revised. (Line 335)

72. Line 288: This sentence is a run-on. Please separate it.

Response: Thank you for pointing out this oversight. The sentence has been revised. (Line 335-338)

73. Line 292: Add a comma after the word "host cells"

Response: Thank you for pointing out this oversight. The text has been revised. (Line 340)

74. Line 298: Please provide references that prove the efficacy of your culturing techniques for producing viable *P. gingivalis* growth.

Response: Thank you for this direction. The references have been included. (Line 348)

75. Line 300: Is there any particular reason why 10% sheep blood was used as

almost all published papers on *P. gingivalis* used 5%. One more thing, it is defibrinated, not defibrated.

Response: Thank you for your thorough review and salient observations. We adopted the medium from the previous papers. We used 5%, 10%, and 20% sheep blood to culture *P. gingivalis*. There wasn't any significantly difference in the bacteria growth. The text and reference have been revised. (Line 349-350)

76. Line 302-303: please be consistent with which one was true, jar or chamber?

Response: Thank you for your thorough review and salient observations. The text has been revised. (Line 350)

77. Line 307: how is it possible to have OD so low about one even after more than 48 h of growth?

Response: Thank you for pointing out this oversight. The text and references have been revised. (Line 356)

78. Line 311: what does mean by (-) in PBS (-).

Response: Thank you for pointing out this oversight. The text has been revised. (Line 360)

79. Line 329: add chronic before periodontitis.

Response: Thank you for pointing out this oversight. The text has been revised. (Line

383)

80. Line 336: How was the nM concentration determined as it needs Molecular Weight (MW) for molar concentration calculation, and it would be interesting to know how was the MW of OMVs determined?

Response: Thank you for pointing out this oversight. The unit has been revised. (Line 390)

81. Line 474: Please attempt to maintain a singular tense. Additionally, the term "should" should not be utilize as it is expected that these methodologies were performed.

Response: Thank you for pointing out this oversight. The text has been revised. (Line 531)

82. Figure 1: The bands of Figure 1D, especially when comparing the bands of the *P. gingivalis* OMVs to free *P. gingivalis*, look very blurred and dissimilar. A repeated picture is necessary.

Response: Thank you for your thorough review and salient observations. The silver staining has been repeated and we have uploaded a coomassie blue staining in the supplementary material (Fig. S1).

Fig. S1 Coomassie Blue Staining of *P. gingivalis* OMVs and *P. gingivalis*.

83. Figure 3: Please increase the intensity of the red in Figure 3A and 3C, as it is very difficult to see the Dil-labelled OMVs and the PI stain, respectively.

Response: Thank you for your thorough review and salient observations. The figures have been revised.

84. Figure 5: Please increase the intensity of the red in Figure 5F, as it is very difficult to see the PI stain.

Response: Thank you for your thorough review and salient observations. The figure has been revised.

85. Figure 6: Please increase the intensity of the red in Figure 6F, as it is very difficult to see the PI stain.

Response: Thank you for your thorough review and salient observations. The figure has been revised.

Once again, thank you very much for your comments and suggestions. We are grateful for the time and energy you expended on our behalf.

Responses to Reviewer #2

Thank you for your detailed comments and suggestions. Considering these suggestions, we tried our best to improve the manuscript and made corresponding revisions.

Major comments:

1. Line no. 67-70, page no. 4, “Indeed, a recent study has revealed that OMVs could deliver a novel class of small RNAs (msRNAs, 15-25 nucleotides) into the host,in the meantime” Please cite the reference. Mention the target of OMVs delivery.

Response: Thank you for pointing out this oversight. The text and reference have been revised in the manuscript. (Line 80)

2. Line no. 305-318, page no. 15-16, “Isolation and identification of *P. gingivalis* OMVs”. Authors have taken the death phase culture of *P. gingivalis* for isolating OMVs. Death phase culture contains high amount of lysed bacterial products/remnants contaminations, which can be found in the culture supernatant even after filtration (0.22- μ m) and ultracentrifugation (100000 \times g). This may effect/ influence the test result.

Response: The authors recognize the reviewer’s concern. We adopted our isolation method based on the established protocols (1–3). Our functional test results are consistent with other reports (2–5). With the reviewer’s notion, we’ll try to isolate OMVs at different growth stages in the future.

Reference:

1. Farrugia C, Stafford GP, Murdoch C. 2020. Porphyromonas gingivalis Outer Membrane Vesicles Increase Vascular Permeability. J Dent Res 002203452094318.
 2. Seyama M, Yoshida K, Yoshida K, Fujiwara N, Ono K, Eguchi T, Kawai H, Guo J, Weng Y, Haoze Y, Uchibe K, Ikegame M, Sasaki A, Nagatsuka H, Okamoto K, Okamura H, Ozaki K. 2020. Outer membrane vesicles of Porphyromonas gingivalis attenuate insulin sensitivity by delivering gingipains to the liver. Biochim Biophys Acta BBA - Mol Basis Dis 1866:165731.
 3. He Y, Shiotsu N, Uchida-Fukuhara Y, Guo J, Weng Y, Ikegame M, Wang Z, Ono K, Kamioka H, Torii Y, Sasaki A, Yoshida K, Okamura H. 2020. Outer membrane vesicles derived from Porphyromonas gingivalis induced cell death with disruption of tight junctions in human lung epithelial cells. Arch Oral Biol 118:104841.
 4. Zhang Z, Liu S, Zhang S, Li Y, Shi X, Liu D, Pan Y. 2022. Porphyromonas gingivalis outer membrane vesicles inhibit the invasion of Fusobacterium nucleatum into oral epithelial cells by downregulating FadA and FomA. J Periodontol 93:515–525.
 5. Liu D, Liu S, Liu J, Miao L, Zhang S, Pan Y. 2021. sRNA23392 packaged by Porphyromonas gingivalis outer membrane vesicles promotes oral squamous cell carcinomas migration and invasion by targeting desmocollin-2. Mol Oral Microbiol 36:182–191.
- 3. Authors are requested to check the OMVs' sterility by culturing on plates.**

Response: The authors recognize the reviewer's concern. The OMVs' sterility has been checked.

4. Please mention the count of OMVs (particles/mL) mixed with cell culture for pretreatment.

Response: The authors appreciate the reviewer pointing out this oversight. The amount of OMVs for cell culture treatment is 10 $\mu\text{g/ml}$. The text has been revised accordingly and we also uploaded the pre-experimental results of *P. gingivalis* OMVs concentration-dependent induction of cell death in the supplementary materials.

Fig. S3 The effect of different concentration of *P. gingivalis* OMVs on the viability of hPDLCs. (A) hPDLCs treated with *P. gingivalis* OMVs (0-50 $\mu\text{g/ml}$) for 24h and viability measured by CCK-8 assay. (B) Representative images taken by light microscopy. Scale bar=20 μm .

5. Fig. 1D, please mention the molecular weight of protein ladder. Also to be mentioned the amount of proteins loaded in each well of SDS-PAGE.

Response: The authors appreciate the reviewer pointing out this oversight. The molecular weight of the protein ladder has been added in Fig. 1D. And the amount of protein has been mentioned in the material and method section. (Line 368-372)

6. Fig. 2D, what is indicating by red arrows? Please mention in the legend.

Response: The authors appreciate the reviewer pointing out this oversight. The figure legend has been revised to clarify this matter. (Line 761-762)

7. Fig. 3I, TNF- α is not showing significant change with OMVs. How can authors explain it? Please write in the discussion.

Response: Thank you for your thorough review and salient observations. The information related to this was added to the Discussion section in the revised manuscript. (Line 284-289)

Minor Comments:

- 1. Fig. 1B, scale is not visible properly. Please mention scale in white.**

Response: The authors appreciate the reviewer pointing out this oversight. The scale was generated when captured by TEM thus it's difficult to change the color. We've mentioned the scale in the figure legends. (Line 753)

- 2. Fig. 2A, E, F, please increase font size and make the fonts more clearly to read.**

Response: The authors appreciate the reviewer pointing out this oversight. We've modified the font size in Figure. 2A, E, F.

- 3. Reference No-34, 45 please rewrite as per the Journal's standard format.**

Response: The authors appreciate the reviewer pointing out this oversight. The format has been revised.

Once again, thank you very much for your comments and suggestions. We are grateful for the time and energy you expended on our behalf.

Responses to Reviewer #3

Thank you for your detailed comments and suggestions. We found them quite useful as we approached our revision. We tried our best to improve the manuscript and made corresponding revisions.

Abstract

Line 24: "hPLDCs" should be "hPDLDCs". Please correct it and revise this throughout the manuscript.

Response: The authors appreciate the reviewer pointing out this oversight. The text has been revised. We have checked and corrected the manuscript. (Line 25-26)

Lines 26-29: The sentence in these lines is confusing (it seems like the sRNA 45033 regulates apoptosis in hPDLDCs using a dual-luciferase reporter assay, overexpression and knockdown methods). Please re-write it.

Response: The authors recognize the reviewer's concern. According to your helpful advice, we've rewrote the sentences and marked the text in yellow color. (Line 30-32)

Introduction:

Line 76: "binding" should be "binds".

Response: Thank you for pointing out this oversight. The text has been revised. (Line 94)

Lines 95-97: The authors mention in line 61 that *P. gingivalis* OMVs have a size of around 100nm. In the results of this manuscript (Figure 1C), the authors report isolation of OMVs with an average of 202.8 +71.87 nm (ranging from 100 to about 500 nm). How do the authors explain that the size of the OMVs isolated by them are greater than the 100nm size previously reported? Is there a difference in OMVs size according to different *P. gingivalis* strains?

Response: The authors recognize the reviewer's concern. We think that the size of the OMVs is varied based on several factors, the strains, the bacterial condition, and the isolation protocol. We hypothesize that different sizes could be important for OMVs' entry into the host cells. However, there aren't many reports regarding this issue in the current field.

We adopted our isolation method based on the established protocols (1–3). The difference in average diameter might be caused by the differences in the isolation protocol. But our results consistent with the ranges that others reported (2–5). We've revised the manuscript to avoid confusion. (Line 66)

Reference:

1. Farrugia C, Stafford GP, Murdoch C. 2020. Porphyromonas gingivalis Outer Membrane Vesicles Increase Vascular Permeability. J Dent Res 002203452094318.
2. Seyama M, Yoshida K, Yoshida K, Fujiwara N, Ono K, Eguchi T, Kawai H, Guo J, Weng Y, Haoze Y, Uchibe K, Ikegame M, Sasaki A, Nagatsuka H, Okamoto K, Okamura H, Ozaki K. 2020. Outer membrane vesicles of Porphyromonas gingivalis attenuate insulin sensitivity by delivering gingipains to the liver. Biochim Biophys Acta BBA - Mol Basis Dis 1866:165731.
3. He Y, Shiotsu N, Uchida-Fukuhara Y, Guo J, Weng Y, Ikegame M, Wang Z, Ono K, Kamioka H, Torii Y, Sasaki A, Yoshida K, Okamura H. 2020. Outer membrane vesicles derived from Porphyromonas gingivalis induced cell death with disruption of tight junctions in human lung epithelial cells. Arch Oral Biol 118:104841.
4. Zhang Z, Liu S, Zhang S, Li Y, Shi X, Liu D, Pan Y. 2022. Porphyromonas gingivalis outer membrane vesicles inhibit the invasion of Fusobacterium nucleatum into oral epithelial cells by downregulating FadA and FomA. J Periodontol 93:515–525.
5. Liu D, Liu S, Liu J, Miao L, Zhang S, Pan Y. 2021. sRNA23392 packaged by

Porphyromonas gingivalis outer membrane vesicles promotes oral squamous cell carcinomas migration and invasion by targeting desmocollin-2. Mol Oral Microbiol 36:182–191.

Line 309: The authors describe that the supernatants from *P. gingivalis* is filtered using a 0.22 µm syringe filter before the samples are centrifuged at high speed to concentrate the OMVs. In figure 1C, the authors show that their OMVs vary in size (ranging from 100 to about 500nm) with an average size of 202.8+71.87 nm. How can the authors obtain OMVs that are larger than the pore of the filter (200nm)? Is it possible that the OMVs fuse after the concentration step? This point should be discussed and added to the manuscript.

Response: The authors recognize the reviewer's concern. OMVs are composed of one single lipid bilayer. OMVs will likely fuse after the ultra-centrifuge or concentration step, and after elution in PBS, the sizes could be also enlarged by osmotic pressure. However, there isn't any evidence in the current field. We've discussed this point in the revised manuscript. (Line 266-268)

Methods: The authors should state, at least when it is first cited, what are hPDLCs. The authors mention that the primary hPDLCs were isolated from human samples. How did the authors confirm that the cells in the manuscript were hPDLCs? Are there cell markers to detected by the authors that confirmed that the cells used in this study were hPDLCs?

Response: Thank you for your detailed comments and suggestions. We've added this matter in the revised manuscript. "hPDLCs" has been defined in the manuscript (Line 25-26). We've uploaded the figures that confirm the hPDLCs in the supplementary materials (1, 2). (Line 411-412)

Fig. S2 Characterization of hPDLCs. (A) Primary tissue culture under light microscopy. (B) Immunohistochemical staining showed negative expression of keratin. (C) Immunohistochemical staining showed positive expression of vimentin.

Reference:

1. Li D-D, Pan J-F, Ji Q-X, Yu X-B, Liu L-S, Li H, Jiao X-J, Wang L. 2016. Characterization and cytocompatibility of thermosensitive hydrogel embedded with chitosan nanoparticles for delivery of bone morphogenetic protein-2 plasmid DNA. *J Mater Sci Mater Med* 27:134.
2. Li Q, Luo T, Lu W, Yi X, Zhao Z, Liu J. 2019. Proteomic analysis of human periodontal ligament cells under hypoxia. *Proteome Sci* 17:3.

Results:

Fig 1A: The authors should add in the image that the chronic periodontitis control group received PBS injections (instead of OMVs). Is that correct to assume?

Response: The authors appreciate the reviewer pointing out this oversight. Figure 2A

has been revised.

What was the volume used to inject PBS or OMVs to rats' mouths?

Response: Thank you for pointing out this oversight. The volume (20µl) has been added in the material and methods section. (Line 390)

Line 113: The authors should explain what "Dil-labeled *P. gingivalis* OMVs" is, in the results section. Or at least refer the readers to the Methods section.

Response: Thank you very much for your valuable suggestion. The text has been revised in the material and methods section. (Line 420-421)

Figure 3: How much of the OMVs were used in the experiments? The authors show a time-dependent induction of cell death by OMVs, but they do not show if OMVs induce cell death in low or high concentrations/amounts. I recommend that the authors add results on the different concentrations/ amounts of OMVs that can induce cell death of hPDLCs

Response: The authors appreciate the reviewer pointing out this oversight. We've mentioned the amount of OMVs (10 µg/ml) in the material and methods section. We also uploaded the pre-experimental results of *P. gingivalis* OMVs concentration-dependent induction of cell death in the supplementary materials.

Fig. S3 The effect of different concentration of *P. gingivalis* OMVs on the viability of hPDLCs. (A) hPDLCs treated with *P. gingivalis* OMVs (0-50 µg/ml) for 24h and viability measured by CCK-8 assay. (B) Representative images taken by light microscopy.

Figure 3A: The authors show, by immunofluorescence microscopy, that there is co-localization of OMVs (in red) with hPDLCs (in green). To this reviewer, this experiment shows co-localization instead of internalization of OMVs. Can the authors show internalization of OMVs by hPDLCs using a different technique? Confocal microscopy? Z-stack images/videos? Flow cytometry? Hence, Figure 3A has an additional text under the word "merge". It is unclear for me what the text means.

Response: Thank you for your thorough review and salient observations. Figure 3A has been re-produced using confocal microscopy.

Figure 3G: The authors should include the densitometric analyses to western blot images in Figure 3G.

Response: The authors appreciate the reviewer pointing out this oversight. We've included densitometric analyses to all western blot results.

Figure 4A and 4D: The data in these images are too small and impossible to read. The authors should provide better quality images, or provide the information as supplementary data, if they don't fit in the main text of the manuscript.

Response: Thank you for your detailed comments and suggestions. Figure 4 has been re-arranged and other information have been provided in Figure S5.

Figure 5: The experiment conducted in Fig5B needs to be better explained in the results section.

Response: Thank you for this instructive suggestion. We have tried to clarify Fig. 5B in the revised manuscript. (Line 194-198)

Are the data on the transcriptome sequencing analysis of *Porphyromonas gingivalis* OMVs available in a public repository?

Response: Thank you for pointing out this oversight. NGS data sets have been deposited into GEO (accession number GSE218606). (Line 179-180)

Once again, thank you very much for your comments and suggestions. We are grateful for the time and energy you expended on our behalf.

Responses to Reviewer #4

Thank you for your thorough review. Considering these suggestions, we tried our best to improve the manuscript and made corresponding revisions.

Major comments:

• Since the OMVs isolated had a broad distribution in terms of size, do the authors think that size differences in OMVs can impact the results in the experiments performed? Additionally, can the author comment on whether OMVs size composition/population differ among biological replicates?

Response: The authors recognize the reviewer's concern. We think that the size of the OMVs is varied based on several factors, the strains, the bacterial condition, and the isolation protocol. We hypothesize that different sizes could be important for OMVs' entry into the host cells or OMVs' content. However, there aren't many reports regarding this issue in the current field. We've discussed this issue in the revised manuscript. (Line 266-268)

• The authors reasoned that OMV-derived sRNA45033 is targeting CBX5. Did the authors consider using OMVs isolated from a mutant strain lacking sRNA45033 as a negative control? Although the use of CBX5 wild-type (pCBX5-WT) and mutant plasmid (pCBX5-Mut), and its inhibitor indeed validated the role of sRNA45033, I wondered to what degree sRNA45033 causes cell apoptosis. Is it possible that there are other redundant sRNA?

Response: The authors recognize the reviewer's concern. It'll be indeed interesting and convenient to have a mutant strain lacking sRNA45033 as a negative control to explore the bio-function of OMVs. Some reports use a gingipain-deficient strain to explore the mechanism of *P. gingivalis* OMVs (1). However, it's a complex process and there is still much controversy about the biogenesis of OMVs and their content (2). Targeting specific sRNAs but not affecting the bacteria is another difficult problem that needs to solve. We'll try to produce stable sRNA mutant strains not only sRNA45033 but also other sRNA candidates using a technique like Crisper-Cas9 in the future. Our current findings only suggest that sRNA45033 causes cell apoptosis. But along the course of exploring other msRNAs we found, there will probably overlap in other sRNAs.

Reference:

1. Farrugia C, Stafford GP, Murdoch C. 2020. *Porphyromonas gingivalis* Outer Membrane Vesicles Increase Vascular Permeability. J Dent Res 002203452094318.
2. Sartorio MG, Pardue EJ, Feldman MF, Haurat MF. 2021. Bacterial Outer Membrane Vesicles: From Discovery to Applications. Annu Rev Microbiol 75:609–630.

• It is interesting that PG-derived sRNA45033 target hPLDCs for apoptosis. Did the authors perform a phylogenetic analysis to determine whether this is a conserved mechanism for PG, or at genus level or in gram negative pathogens? I would recommend including this analysis into the main figures.

Response: Thank you very much for your valuable suggestion. Due to the short sequence of msRNAs, we find it difficult to conduct a phylogenetic analysis. But in the

thought of this notion, we've acquired *P. gingivalis* strain W83 and *A. actinomycetemcomitans* to preliminary test this theory in the latter research.

• For Fig 1, did the authors perform normalization and/or separation of OMV based on size? It would be beneficial to show the distribution/probability of the size of OMV harvested from different biological samples to account for biological variability. Assuming the author is making a statement on protein content between OMV and *P. gingivalis*, the OMV and PG cell lysate has to be normalized by protein content which is not seen in Fig 1D. My recommendation is to remove Fig 1D or move to supplementary.

Response: The authors recognize the reviewer's concern. The text in material and methods section has been revised to explain the silver staining procedure in detail. (Line 368-372) In this case, the concentrations of OMVs and PG were measured by BCA assay. The silver staining was utilized to see the corresponding bands between extracted OMVs and PG (1, 2). We also uploaded a Coomassie blue staining from a different batch of extracted OMVs in the supplementary material.

Fig. S1 Coomassie Blue Staining of *P. gingivalis* OMVs and *P. gingivalis*.

Reference:

1. Seyama M, Yoshida K, Yoshida K, Fujiwara N, Ono K, Eguchi T, Kawai H, Guo J, Weng Y, Haoze Y, Uchibe K, Ikegame M, Sasaki A, Nagatsuka H, Okamoto K, Okamura H, Ozaki K. 2020. Outer membrane vesicles of Porphyromonas gingivalis attenuate insulin sensitivity by delivering gingipains to the liver. *Biochim Biophys Acta BBA - Mol Basis Dis* 1866:165731.
2. He Y, Shiotsu N, Uchida-Fukuhara Y, Guo J, Weng Y, Ikegame M, Wang Z, Ono K, Kamioka H, Torii Y, Sasaki A, Yoshida K, Okamura H. 2020. Outer membrane vesicles derived from Porphyromonas gingivalis induced cell death with disruption of tight junctions in human lung epithelial cells. *Arch Oral Biol* 118:104841.

• In Fig 2, the authors reasoned that the hematoxylin (staining nuclei cell purple) and eosin (staining extracellular matrix and cytoplasm pink) stained images support the notion that OMV altered bone resorption, which I do not agree. H&E staining is relatively crude and is not specific to a cell lineage. Further, the H&E image is not quantitative data, but qualitative hence it is not possible to demonstrate significance. Similarly, TRAP stains tartrate-resistant acid phosphatase which is present in osteoclast; however, these enzymes are also present in immune cells too. I would suggest the authors rephrase their interpretations of the result. Alternatively, performing flow cytometry with specific cell markers should suffice to validate their reasoning.

Response: We appreciate the Reviewer's questions and suggestions about this, the interpretation has been revised in the text and figure legends accordingly. H&E staining and TRAP staining were used as assisting evidence to confirm the μ CT analysis (which was biologically meaningful and the statistical analysis in multiple experiments showed that the changes are reproducible and significant). We have tried to clarify that in the revised manuscript (line 131-134). We also attached some references using these techniques to characterize the bone resorption (1, 2).

Reference:

1. Chen Y, Yang Q, Lv C, Chen Y, Zhao W, Li W, Chen H, Wang H, Sun W, Yuan H. 2021. NLRP3 regulates alveolar bone loss in ligature-induced periodontitis by promoting osteoclastic differentiation. *Cell Prolif* 54:e12973.
2. Wang H, Chen Y, Li W, Sun L, Chen H, Yang Q, Zhang H, Zhang W, Yuan H,

Zhang H, Xing L, Sun W. 2020. Effect of VEGFC on lymph flow and inflammation-induced alveolar bone loss. *J Pathol* 251:323–335.

Minor comments:

- **Throughout the manuscript, please italicize *in vivo* and *in vitro*.**

Response: The authors appreciate the reviewer pointing out this oversight. The manuscript has been revised.

- **The first use of the abbreviation hPLDCs appeared in the abstract, yet there is no mention what is hPLDCs. Please clarify the rationale for using hPLDCs.**

Response: The authors appreciate the reviewer pointing out this oversight. “hPDLCs” has been defined in the abstract (Line 25-26). hPDLCs are very similar to the periodontium. The periodontal ligament is a part of the attachment apparatus comprised of periodontal ligament cells, extracellular matrix and fibres, attaching the root cement to the alveolar bone. hPDLCs are in close proximity to bacteria within the plaque and the pocket, thus hPDLCs serves as a good experimental model in our study (1). We also include the reference to clarify this matter. (Line 409)

Reference:

1. Jönsson D, Nebel D, Bratthall G, Nilsson B-O. 2011. The human periodontal ligament cell: a fibroblast-like cell acting as an immune cell. *J Periodontal Res* 46:153–157.

- **In line 68, please reference the study.**

Response: The authors appreciate the reviewer pointing out this oversight. The text has been revised and reference was included. (Line 80)

- **In line 71, please provide some information on the downstream genes regulated by *P. gingivalis* msRNA.**

Response: Thank you for this direction. The information has been included in the revised manuscript. (Line 81-86)

- **In line 76, change "binding" to "bind".**

Response: Thank you for pointing out this oversight. The text has been revised. (Line 94)

- **In line 77, change "trimethylated" to "tri-methylated"**

Response: Thank you for pointing out this oversight. The text has been revised. (Line 95)

- **In line 79, please clarify on the specific function of CBX5.**

Response: Thank you for this direction. The information has been included in the revised manuscript. (Line 98-99)

- **In line 148, "gene regulation" was mentioned twice.**

Response: Thank you for pointing out this oversight. The text has been revised. (Line 178)

- **Fig 4A-F images are too small to be seen. I would recommend moving 4B-D to supplementary, and combine 4E-F together into a bigger figure. This would make room for 4A to be large too. Please provide more information on these figures in the figure legend.**

Response: Thank you very much for your valuable suggestion. Fig 4 has been re-arranged. Other information has been uploaded as supplementary in Fig S5.

- **Please clarify what analysis was performed to determine CBX5 quantification shown in Fig 4K, and what staining are done in Fig 4J.**

Response: The authors appreciate the reviewer pointing out this oversight. The figure legend has been revised to clarify this issue. (Line 785-787)

- **Please clarify the sentence in line 164 "The s45033 mimics significantly decreased the luciferase activity of the wild-type (WT) while did not affect the mutant". I would suggest renaming it to s45033 recombinant WT (r-WT) as the sentence sounds confusing.**

Response: Thank you for this direction. We've tried to clarify this matter in the revised manuscript. (Line 194-198)

• In line 168, the authors mentioned "inhibition of s45033 dramatically reversed 168 cell viability". Can the authors explain the mechanism of inhibition that is happening here?

Response: Thank you for your thorough review and salient observations. The inhibition was caused by the s45033 inhibitor which was a reverse complement to the s45033 mimic. The material and methods section has been revised to clarify this matter. (Line 544-545)

Once again, thank you very much for your comments and suggestions. We are grateful for the time and energy you expended on our behalf.

December 7, 2022

Dr. yan xu
Department of Periodontics, Affiliated Hospital of Stomatology, Nanjing Medical University,
136 Hanzhong Road, Nanjing
nanjing
China

Re: Spectrum03288-22R1 (*Porphyromonas gingivalis* outer membrane vesicles promote apoptosis via msRNA-regulated DNA methylation in periodontitis)

Dear Dr. yan xu:

Your manuscript has been accepted, and I am forwarding it to the ASM Journals Department for publication. You will be notified when your proofs are ready to be viewed.

Sincerely,

John Atack
Editor, Microbiology Spectrum
